# 🌵 Cactus: Accelerating Auto-Regressive Decoding with Constrained Acceptance Speculative Sampling

**Yongchang Hao**$^{\diamond}$    **Lili Mou**$^{\diamond\spadesuit}$
$^{\diamond}$Dept. Computing Science & Alberta Machine Intelligence Institute (Amii), University of Alberta
$^{\spadesuit}$Canada CIFAR AI Chair
yongcha1@ualberta.ca    doublepower.mou@gmail.com

## Abstract

Speculative sampling (SpS) has been successful in accelerating the decoding throughput of auto-regressive large language models by leveraging smaller draft models. SpS strictly enforces the generated distribution to match that of the verifier LLM. This is unnecessarily restrictive as slight variations of the verifier's distribution, such as sampling with top-$k$ or temperature, would also be acceptable. Typical acceptance sampling (TAS) alleviates this issue by accepting more tokens using entropy-based heuristics. However, this approach distorts the verifier distribution, potentially degrading output quality when the verifier encodes critical information. In this work, we formalize the speculative sampling algorithm through the lens of constrained optimization. Based on this formulation, we propose **Cactus** (**c**onstrained **ac**cep**t**ance spec**u**lative **s**ampling), a method that guarantees controlled divergence from the verifier distribution and increasing acceptance rates. Empirical results across a wide range of benchmarks confirm the effectiveness of our approach. The code is publicly available.[1]

## 1 Introduction

Auto-regressive large language models (LLMs) have driven remarkable advances in machine learning and artificial intelligence (Vaswani et al., 2017; Brown et al., 2020; Kaplan et al., 2020), yet their growing size comes with steep computational costs: generating each token requires a memory-bound forward pass through hundreds of billions of parameters, which bottlenecks LLM throughput (Yuan et al., 2024). Speculative sampling (SpS) addresses this by first using a smaller draft model to propose a certain number of candidate tokens autoregressively, then verifying the candidate tokens in parallel with the large-scale *verifier* LLM (Stern et al., 2018; Xia et al., 2023; Leviathan et al., 2023; Chen et al., 2023). Since SpS can emit multiple tokens per large-model invocation, it substantially speeds up auto-regressive generation by alleviating the memory-bound issue.

Despite its success, SpS enforces strict distributional equivalence with the verifier, causing correct but lower-probability tokens to be rejected. In real-world applications, exact adherence to the original distribution is generally not required (Holtzman et al., 2020; Meister et al., 2020). Typical acceptance sampling (TAS; Cai et al., 2024) mitigates this issue by accepting proposals based on entropy-driven heuristics (Hewitt et al., 2022; Meister et al., 2023). However, we show in this paper that TAS improves acceptance rates at the cost of distorting the verifier's output distribution and risking semantic drift when the verifier encodes critical information.

In this work, we reformulate speculative sampling as a constrained optimization problem, explicitly trading off acceptance rate against divergence from the verifier's distribution. Guided by this theory, we introduce Cactus (**c**onstrained **ac**cep**t**ance spec**u**lative **s**ampling), a simple yet principled modification that enforces a hard constraint on distributional divergence while enabling higher acceptance rates.

---

[1] https://github.com/MANGA-UOFA/Cactus

---

**Algorithm 1** Generalized formulation of speculative sampling

---

**Require:** sampling steps $m$, draft model $p$, acceptance rate $\phi$, and recover distribution $g$
1:   $t \leftarrow 1, \boldsymbol{x}_{<t} \leftarrow [\text{BOS}]$
2:   **while** not end **do**
3:      $\triangleright$ Drafting $m$ tokens
4:      **for** $i \leftarrow 0, \ldots, m-1$ **do**
5:        $x_{t+i} \sim p(\cdot|\boldsymbol{x}_{<t+i})$          $\triangleright \boldsymbol{x}_{<t+i}$ is concatenation of $\boldsymbol{x}_{<t}$ and $[x_t, \ldots, x_{t+i-1}]$
6:        $u_i \sim U(0, 1)$          $\triangleright U(0, 1)$ is the uniform distribution between $[0, 1]$
7:      **end for**
8:      $c \leftarrow \min\{j : u_j > \phi(x_{t+j}|\boldsymbol{x}_{<t+j})\} \bigcup\{m\}$      $\triangleright c$ is the length of accepted draft tokens
9:      $x_{t+c} \sim g(\cdot|\boldsymbol{x}_{<t+c})$          $\triangleright x_{t+c}$ is sampled from the recover distribution
10:    $t \leftarrow t + c + 1$
11: **end while**

---

We conducted experiments on a wide range of benchmarks with multiple state-of-the-art large language models. Results show that Cactus consistently improves generation throughput compared with the lossless SpS. In addition, Cactus preserves the generation quality and diversity of the verifier model, due to its explicit divergence constraint.

## 2   APPROACH

We first provide a generalized formulation for speculative sampling. This enables a theoretical analysis of speculative sampling under a constrained optimization framework. Based on this analysis, we propose a new algorithm, Cactus, which provably approximates the verifier distribution $q$ while achieving higher acceptance rates.

### 2.1   GENERALIZATION OF SPECULATIVE SAMPLING

**Speculative sampling.** The vanilla speculative sampling (SpS; Chen et al., 2023) uses a *draft model* $p(\cdot|\boldsymbol{x}_{<t})$ that has a significantly smaller memory footprint than the *verifier model* $q(\cdot|\boldsymbol{x}_{<t})$. At a time step $t$, SpS repeatedly samples $m \in \mathbb{N}_+$ tokens $x_t, \ldots, x_{t+m-1}$ from $p$ in an auto-regressive manner. Then the verifier evaluates the probabilities of the tokens and calculates the *acceptance rate* $\phi(x_{t+i}|\boldsymbol{x}_{<t+i}) = \min\{1, q(x_{t+i}|\boldsymbol{x}_{<t+i})/p(x_{t+i}|\boldsymbol{x}_{<t+i})\}$ for all $i \in [0, m)$. If any token $x_{t+j}$ is rejected, then tokens $x_{t+j+1}, \ldots, x_{t+m-1}$ are also discarded. As a backup, SpS resamples $x_{t+j}$ using the *recover distribution* $g(x_{t+j}|\boldsymbol{x}_{<t+j}) \propto (q(\cdot|\boldsymbol{x}_{<t+j}) - p(\cdot|\boldsymbol{x}_{<t+j}))_+$, where $(\cdot)_+$ denotes $\max(0, \cdot)$. The final accepted tokens are $x_t, \ldots, x_{t+j}$. By this draft-and-verify scheme, SpS accelerates auto-regressive decoding by avoiding the need to load the large verifier model $q$ from memory at every step. This approach has been shown effective in practice (Zhou et al., 2024; Hu et al., 2025).

We formalize the draft-and-verify scheme as Algorithm 1. Under this setting, we can show that the vanilla SpS algorithm (Chen et al., 2023) produces any target distribution with an optimal acceptance rate.

**Observation 1.** *Consider any desired target distribution $h$ and draft model $p$. Algorithm 1 produces the target distribution $h$ exactly if the acceptance rate and recover distribution are defined as*

$$\phi(x_t|\boldsymbol{x}_{<t}) = \min\left\{\frac{h(x_t|\boldsymbol{x}_{<t})}{p(x_t|\boldsymbol{x}_{<t})}, 1\right\} \tag{1}$$

$$\text{and} \quad g(x_t|\boldsymbol{x}_{<t}) = \frac{h(x_t|\boldsymbol{x}_{<t}) - p(x_t|\boldsymbol{x}_{<t})\phi(x_t|\boldsymbol{x}_{<t})}{\mathbb{E}_{x' \sim p}[1 - \phi(x'|\boldsymbol{x}_{<t})]}, \tag{2}$$

*respectively. In addition, this acceptance rate $\phi$ is optimal for achieving the highest acceptance rate.*

*Proof.* See Appendix A.1.      $\square$

## 2.2 APPROXIMATING SPS AS CONSTRAINED OPTIMIZATION

Observation 1 provides a foundation to produce an arbitrary target distribution with the optimal design under the draft-and-verify scheme. Our insight is that, instead of producing a fixed verifier distribution $q$ as the target distribution $h$ like SpS, we may utilize this observation to dynamically select a distribution $h$ close to $q$ while yielding higher acceptance rates based on function $\phi$. This can be formulated as a constrained optimization problem.

For each step $t$, assume the drafted token has index $n$. Let $\boldsymbol{h} \in \mathbb{R}^{|V|}$ be the parameters to be optimized. The ideal $h$ is given by $h(i|\boldsymbol{x}_{<t}) = \boldsymbol{h}_i^*$, where $\boldsymbol{h}^*$ is the solution of the following problem:

$$\max_{\boldsymbol{h}} \quad \min\{h_n/p(n|\boldsymbol{x}_{<t}), 1\} \tag{3}$$

$$\text{s.t. } \boldsymbol{h} \in \Delta^{|V|-1} \tag{4}$$

$$D_f(\boldsymbol{h}\|q(\cdot|\boldsymbol{x}_{<t})) \leq \delta. \tag{5}$$

Here, the hyper-parameter $\delta \geq 0$ controls the closeness to the verifier model $q$, and $D_f$ is any $f$-divergence metric used to measure the distance between $q$ and $h$. Once the optimal $\boldsymbol{h}$ is found, we can then derive the corresponding $\phi$ and $g$ by Observation 1.

The above formulation falls into the framework of constrained convex optimization, which we show has the following solution.

**Theorem 2.** *The optimal solution of $\boldsymbol{h}$ in objective* (3) *is*

$$h_i = \begin{cases} \gamma^*, & \text{if } i = n, \\ \frac{1-\gamma^*}{1-q(n|\boldsymbol{x}_{<t})} q(i|\boldsymbol{x}_{<t}), & \text{otherwise}, \end{cases} \tag{6}$$

*where $\gamma^*$ is any root of the equation*

$$\delta = q(n|\boldsymbol{x}_{<t})f\left(\frac{\gamma}{q(n|\boldsymbol{x}_{<t})}\right) + (1 - q(n|\boldsymbol{x}_{<t}))f\left(\frac{1-\gamma}{1-q(n|\boldsymbol{x}_{<t})}\right) \tag{7}$$

*over the interval $[q(n|\boldsymbol{x}_{<t}), +\infty)$, clamped into $[q(n|\boldsymbol{x}_{<t}), 1]$. The function $f$ is the one used in the definition of $f$-divergence.*

*Proof.* See Appendix A.2. □

Theorem 2 theoretically characterizes the trade-off between closeness to the verifier model $q$ and the acceptance rate induced by $\phi$. In particular, the theorem suggests that the drafted token now has at least the same or a higher chance of being accepted (since $\gamma^* \geq q_n$). For other non-sampled tokens, their probabilities are scaled down proportionally so that $h$ remains a valid distribution.

It is worth noting that, since the solved $\boldsymbol{h}$ in Equation (6) depends on the sampled token $n$, the solution is different for different sampled tokens. As a result, the effective distribution of the overall algorithm $\boldsymbol{h}_{\text{alg}}$ might have a divergence other than $\delta$ from the target distribution $q$. To this end, we provide the following theorem to guarantee the controlled divergence of the effective distribution.

**Theorem 3.** *Let $\phi_n$ and $g_n$ denote the functions that follow the solution in Theorem 2 when the sampled token is $n$. The distribution of the overall algorithm is given by*

$$\boldsymbol{h}_{alg} = \sum_{n \in [|V|]} p(n|\boldsymbol{x}_{<t}) \left[\phi_n(n)\boldsymbol{e}_n + (1 - \phi_n(n))\boldsymbol{g}_n\right], \tag{8}$$

*where $\boldsymbol{e}_n$ is a one-hot vector with only non-zero element at index $n$. In addition,*

$$D_f(\boldsymbol{h}_{alg}\|q(\cdot|\boldsymbol{x}_{<t})) \leq \min\{\Gamma(\delta), D_f(p(\cdot|\boldsymbol{x}_{<t})\|q(\cdot|\boldsymbol{x}_{<t}))\} \tag{9}$$

*for any $\delta \geq 0$. Here, the function $\Gamma : [0, +\infty) \to [0, +\infty]$ is continuous and non-decreasing in $\delta$ with a value of 0 at $\delta = 0$.*

*Proof.* See Appendix A.3. □

In essence, despite the $\boldsymbol{h}$ in Equation (6) is solved specifically for the sampled token $n$, the divergence between the overall distribution and the target distribution is still implicitly controlled. In particular, for any target divergence $0 \leq \delta_{\text{alg}} < +\infty$ imposed on the overall algorithm, we can always find a proper $\delta \geq 0$ such that $D_f(\boldsymbol{h}_{\text{alg}} \| q) \leq \Gamma(\delta) \leq \delta_{\text{alg}}$. While $\Gamma$ does not admit a closed-form expression, $\delta$ itself is a hyper-parameter. In practice, one can tune $\delta$ to achieve the desired quality-throughput trade-off. This confirms the soundness of our framework.

In fact, our framework also offers a novel theoretical interpretation of typical acceptance sampling.

**Proposition 4.** *Typical acceptance sampling (TAS; Cai et al., 2024) implicitly solves a variant of the optimization problem in objective* (3)*, where the $f$-divergence is substituted with the cross-entropy $H(\boldsymbol{h}, q(\cdot | \boldsymbol{x}_{<t}))$.*

*Proof.* See Appendix A.4. □

The suboptimality of TAS arises from the nature of cross-entropy. Specifically, the cross-entropy can be decomposed as

$$H(\boldsymbol{h}, q(\cdot | \boldsymbol{x}_{<t})) = \underbrace{D_{\text{KL}}(\boldsymbol{h} \| q(\cdot | \boldsymbol{x}_{<t}))}_{\text{Mode capturing}} + \underbrace{H(\boldsymbol{h})}_{\text{Certainty}}. \tag{10}$$

Here, the KL divergence encourages $h$ to focus on the mode of $q$ (since $h$ is the first argument), while the entropy term encourages $h$ to be deterministic. However, the summation allows $h$ to collapse into a deterministic distribution at the expense of increasing divergence, therefore failing to capture the full shape of $q$. In fact, TAS always yields $h$ with entropy 0 while increasing the divergence by at least $H(q)$. As a result, the produced distribution may diverge significantly from the verifier model, especially when $q$ carries high entropy and thus rich information.

## 2.3 CACTUS: CONSTRAINED ACCEPTANCE SPECULATIVE SAMPLING

Based on our analysis above, we propose to use only the KL divergence as the measure of "distance". Specifically, this corresponds to the function $f(t) = t \log t$. Combined with our Theorem 2, $\gamma^*$ is the root of

$$\Phi(\gamma) := q(n|\boldsymbol{x}_{<t}) f\left(\frac{\gamma}{q(n|\boldsymbol{x}_{<t})}\right) + (1 - q(n|\boldsymbol{x}_{<t})) f\left(\frac{1-\gamma}{1 - q(n|\boldsymbol{x}_{<t})}\right) \tag{11}$$

$$= \gamma \log\left(\frac{\gamma}{q(n|\boldsymbol{x}_{<t})}\right) + (1-\gamma) \log\left(\frac{1-\gamma}{1 - q(n|\boldsymbol{x}_{<t})}\right) \tag{12}$$

$$= \delta. \tag{13}$$

However, since $\Phi$ is a transcendental function involving terms like $x \log x$, it cannot be solved in closed form. We therefore approximate $\Phi$ by its second-order Taylor series expanded at $\gamma_0 = q(n|\boldsymbol{x}_{<t})$:

$$\Phi(\gamma) \approx \Phi(\gamma_0) + \Phi'(\gamma_0)(\gamma - \gamma_0) + \frac{\Phi''(\gamma_0)}{2}(\gamma - \gamma_0)^2. \tag{14}$$

This approximation is justified by noting that $\delta$ is typically small and $\gamma^*$ remains close to $q(n|\boldsymbol{x}_{<t})$.

**Corollary 5** (Cactus's solution). *Let the $f$-divergence in objective* (3) *be the KL divergence. The solution to Equation* (14) *is given by*

$$h(i|\boldsymbol{x}_{<t}) = \begin{cases} \gamma^*, & \text{if } i = n, \\ \frac{1-\gamma^*}{1-q(n|\boldsymbol{x}_{<t})} q(i|\boldsymbol{x}_{<t}), & \text{otherwise,} \end{cases} \tag{15}$$

*where $\gamma^* = \min\left\{ q(n|\boldsymbol{x}_{<t}) + \sqrt{2\delta q(n|\boldsymbol{x}_{<t})(1 - q(n|\boldsymbol{x}_{<t}))}, 1 \right\}$.*

*Proof.* See Appendix A.5 □

In other words, Cactus modifies the distribution of the verifier model by increasing the probability of the candidate token $n$ by a small "bonus" determined jointly by $q(n|\boldsymbol{x}_{<t})$ and $\delta$. We further show that Cactus's solution is more conservative than the exact solution when the verifier is less confident, ensuring that it strictly satisfies the divergence constraint in such cases.

**Corollary 6.** *When the exact solution $\gamma^*$ is not greater than $0.5$ (i.e., the token is not likely to be accepted), our approximation always satisfies the divergence constraint:*

$$D_{KL}(h\|q) \leq \delta, \tag{16}$$

*where $h(n|\boldsymbol{x}_{<t})$ is given by the approximated solution in Equation (15).*

*Proof.* See Appendix A.6. □

It is easy to see that the bonus probability attains its maximum when $q(n|\boldsymbol{x}_{<t}) = 0.5$. In practice, LLMs generally have more than 100K tokens (Dubey et al., 2024; Qwen Team, 2024), so a probability around $0.5$ indicates strong model confidence in the token. However, SpS could still reject the token $n$ solely because the draft model is overconfident (i.e., $p(n|\boldsymbol{x}_{<t})$ is large). Cactus increases the acceptance likelihood in such scenarios by modifying the verifier distribution accordingly.

Compared with TAS's criterion function, Cactus only requires reading the probability at token $n$ rather than accessing the full vocabulary. This allows Cactus to further reduce memory access overhead, especially in large-vocabulary settings. More importantly, Cactus 's divergence is tightly controlled with minimal entropy change, whereas TAS yields only low-entropy solutions.

## 3 EXPERIMENTS

### 3.1 SETTINGS

**Datasets.** We evaluated Cactus on three popular benchmark datasets for large language models: (1) The **GSM8K** (Cobbe et al., 2021) test set contains 1.3K high-quality grade school math word problems, designed to assess a model's ability to apply mathematics to real-world scenarios. Following common practice in LM-Eval (Gao et al., 2024), we used 5-shot examples for each test instance. The final accuracy score is averaged over all samples. (2) The **IFEval** (Zhou et al., 2023) benchmark measures instruction-following ability. It consists of 500 "verifiable instructions" whose outputs can be heuristically evaluated. For example, a prompt might be: "Write a blog post with 400 or more words about the benefits of sleeping in a hammock," which can be automatically checked by counting the number of words. (3) The **GPQA** (Rein et al., 2024) diamond benchmark includes approximately 200 challenging science questions authored by domain experts, designed to test models' scientific knowledge. For instance, a sample question is: "The angular size of the event horizon of a supermassive black hole in the centre of a galaxy at a distance of $d = 10^{10}$ parsecs is measured to be $\theta = 10^{-17}$ degrees. Find the order of magnitude of the entropy of the black hole." Following common practice (Gao et al., 2024), we format the prompts to include four multiple-choice options. Models are then evaluated by generating a chain-of-thought (Wei et al., 2022) followed by the final answer.

**Evaluation metrics.** For all three tasks, the results are extracted from the generated text by regex matching with the corresponding format. These results are then compared with the gold labels using strict-match accuracy (i.e., 1 if the strings are identical and 0 otherwise). Final scores are obtained by averaging the accuracies over all samples. Following previous work (Dubey et al., 2024), the regex for GSM8K and GPQA is the "flexible-extract" pattern, which selects the first number in the generated sentence regardless of whether the model adheres to the few-shot examples. For IFEval, we use the "prompt-level-strict-acc" regex as defined in Qwen Team (2024), which requires the model to strictly follow all the instructions.

In addition to task scores, we report the average acceptance length (AL) for all runs. Specifically, $\text{AL}_m$ refers to the expected number of accepted tokens among $m$ drafted tokens. A higher $\text{AL}_m$ generally indicates faster generation. However, a method may artificially inflate AL by accepting low-quality draft tokens that lead to longer, less efficient chains of thought. Although AL remains

Table 1: The results on three benchmarks: GSM8K, IFEval, and GPQA. We report the "strict-match" accuracy as the score with the standard regex pattern for each task. $AL_m$ indicates the number of accepted tokens when the draft length is $m$. Rej denotes the total number of rejected tokens throughout generation in relative scale, where we use the SpS runs as the reference (labeled as "Ref").

(a) The results of Qwen 3 8B as verifier and Qwen 3 0.6B as drafter.

| $m$ | Name | GSM8K | | | IFEval | | | GPQA | | |
|---|---|---|---|---|---|---|---|---|---|---|
| | | $Score^\uparrow$ | $AL_m^\uparrow$ | $Rej^\downarrow$ | $Score^\uparrow$ | $AL_m^\uparrow$ | $Rej^\downarrow$ | $Score^\uparrow$ | $AL_m^\uparrow$ | $Rej^\downarrow$ |
| | Verifier | $84.31_{\pm0.47}$ - | | - | $84.66_{\pm0.56}$ - | | - | $41.07_{\pm1.77}$ - | | - |
| 10 | SpS | 83.78 | 4.49 | Ref | 84.66 | 2.59 | Ref | 40.91 | 3.70 | Ref |
| | TAS | 86.58 | 5.49 | -29% | 85.40 | 3.28 | -27% | 41.41 | 5.17 | -42% |
| | Cactus 0.75 | 85.97 | 5.65 | -34% | 85.03 | 3.40 | -31% | 41.42 | 5.33 | -47% |
| | Cactus 1.0 | 86.35 | 5.72 | -37% | 84.10 | 3.44 | -32% | 39.39 | 5.44 | -48% |
| 20 | SpS | 84.46 | 5.44 | Ref | 84.10 | 2.74 | Ref | 42.93 | 4.23 | Ref |
| | TAS | 85.51 | 7.23 | -35% | 84.10 | 3.77 | -29% | 38.89 | 6.68 | -46% |
| | Cactus 0.75 | 86.66 | 7.50 | -37% | 85.95 | 3.76 | -30% | 40.01 | 6.89 | -47% |
| | Cactus 1.0 | 86.43 | 7.61 | -39% | 84.84 | 4.05 | -33% | 39.90 | 7.05 | -49% |

(b) The results of Qwen 3 14B as verifier and Qwen 3 0.6B as drafter.

| $m$ | Name | GSM8K | | | IFEval | | | GPQA | | |
|---|---|---|---|---|---|---|---|---|---|---|
| | | $Score^\uparrow$ | $AL_m^\uparrow$ | $Rej^\downarrow$ | $Score^\uparrow$ | $AL_m^\uparrow$ | $Rej^\downarrow$ | $Score^\uparrow$ | $AL_m^\uparrow$ | $Rej^\downarrow$ |
| | Verifier | $91.71_{\pm0.52}$ - | | - | $85.09_{\pm0.66}$ - | | - | $40.07_{\pm0.77}$ - | | - |
| 10 | SpS | 91.12 | 4.27 | Ref | 85.03 | 2.19 | Ref | 39.39 | 3.37 | Ref |
| | TAS | 92.65 | 5.24 | -30% | 86.14 | 3.00 | -25% | 38.89 | 4.99 | -46% |
| | Cactus 0.75 | 92.12 | 5.35 | -31% | 86.87 | 3.04 | -29% | 44.95 | 5.14 | -50% |
| | Cactus 1.0 | 93.10 | 5.44 | -32% | 85.96 | 3.03 | -30% | 43.43 | 5.16 | -51% |
| 20 | SpS | 91.89 | 5.11 | Ref | 84.47 | 2.27 | Ref | 40.91 | 3.84 | Ref |
| | TAS | 92.87 | 6.78 | -32% | 85.03 | 3.49 | -27% | 40.40 | 6.41 | -46% |
| | Cactus 0.75 | 92.15 | 7.15 | -36% | 86.69 | 3.45 | -30% | 45.46 | 6.46 | -46% |
| | Cactus 1.0 | 92.87 | 7.00 | -34% | 86.32 | 3.60 | -30% | 45.46 | 6.74 | -50% |

high in this case, the behavior can lead to lower overall throughput due to unnecessarily lengthy outputs. To present a more complete picture of generation efficiency, we also measure the number of rejected tokens during generation, which reflects both the acceptance rate and the total length of generation.

**Implementation details.** We used the Qwen 3 series as our main testbed for two reasons: (1) the models come in a variety of sizes, ranging from 0.6B to 14B parameters, enabling a wide range of choices of model pairs; (2) the models are trained to generate with internalized chain-of-thought reasoning (Wei et al., 2022), which makes them a natural use case for speculative sampling given the longer generation lengths (Yang et al., 2025b). For all experiments, we used the recommended generation parameters (Yang et al., 2025a), where top-$p$ is set to 0.95, top-$k$ is set to 20, and temperature is 0.6.

## 3.2 Main results

As shown in Table 1, speculative sampling (SpS) serves as a strong baseline that closely preserves the output distribution of the verifier model. Across all three benchmarks (GSM8K, IFEval, and GPQA), SpS maintains similar accuracies to the verifier (e.g., 84.46 vs. 84.31 on GSM8K with $m = 20$ in Table 1a, and 91.89 vs. 91.71 in Table 1b). This aligns with the theoretical claim that SpS is nearly lossless in generation quality. Additionally, the number of accepted tokens ($AL_m$) for SpS reaches 5.44 on GSM8K and 4.23 on GPQA with $m = 20$, indicating that the verifier model is invoked less frequently.

Figure 1: Accuracy-acceptance across benchmarks and model settings. The $x$-axis shows the average accepted length (AL), and the $y$-axis shows accuracy normalized by the standard deviation from the verifier.

Typical acceptance sampling (TAS) outperforms SpS in terms of acceptance rate, achieving more accepted tokens and lower rejection rates. For example, on GSM8K with $m = 20$, TAS improves $AL_m$ from 5.44 to 7.23 (Table 1a) and reduces the rejection rate by 35%, which is consistent with our approximation analysis in Section 2.2. However, TAS often introduces distributional shifts that degrade performance. For instance, on GPQA in Table 1a, TAS yields lower accuracy than SpS (38.89 vs. 42.93), likely due to accepting plausible yet suboptimal tokens, especially when the verifier distribution contains fine-grained decision signals.

By contrast, our proposed method, Cactus, achieves the highest acceptance rates across all benchmarks while maintaining or improving accuracy. When $\delta = 0.75$, Cactus consistently surpasses both SpS and TAS in $AL_m$ while achieving strong accuracy, e.g., a score of 86.66 on GSM8K with $m = 20$ (Table 1a) and 45.46 on GPQA with $m = 20$ (Table 1b), notably outperforming all baselines. When $\delta = 1.0$, Cactus further increases $AL_m$ to 7.61 on GSM8K with a score of 86.43 (Table 1a), or to 7.00 with a score of 92.87 using a larger verifier (Table 1b). Notably, unlike TAS, Cactus does not degrade performance on challenging benchmarks such as GPQA. Instead, it achieves both high acceptance rates and stable accuracy, validating its theoretical foundation in constrained optimization and demonstrating practical robustness across diverse tasks.

## 3.3 IN-DEPTH ANALYSES

**Accuracy against acceptance rates.** We visualize the accuracy-acceptance trade-off in Figure 1, where accuracy is measured in standard deviations ($\sigma$) from the verifier mean, and throughput is quantified by the average accepted length (AL). Each subplot corresponds to a specific benchmark and verifier-drafter pair. The dashed black line indicates the verifier performance, and the red dashed line marks the $-1\sigma$ threshold, a commonly used indicator of statistically significant degradation.

As shown, TAS improves throughput over SpS but often suffers from accuracy drops, notably falling below the $-1\sigma$ threshold on GPQA with the 8B verifier ($m = 20$). In contrast, Cactus consistently preserves accuracy (remaining within or above the verifier confidence range) and frequently exceeds it, such as on GSM8K and IFEval with both 8B and 14B verifiers. This demonstrates that Cactus effectively improves decoding efficiency without compromising (and sometimes even enhancing) generation quality.

It is also worth noting that the improvements from Cactus are stable across tasks with different characteristics. For instance, on challenging benchmarks like GPQA, where other methods either exhibit significant degradation (e.g., TAS) or achieve limited throughput gains (e.g., SpS), Cactus substantially increases AL while maintaining accuracy above baseline. This highlights the strength of our constrained acceptance framework in balancing aggressive token acceptance with distributional fidelity.

**The importance of divergence control.** Our Cactus dynamically manipulates the target distribution to increase the chance of accepting the sampled tokens. Since this inevitably pushes the target distribution $h$ from the verifier $q$ to be more similar to the draft model $p$, it resembles the notion of mixing the distributions $q$ and $p$ by interpolation. However, we argue that Cactus is superior to simple interpolation, given that it uses a principled approach which comes with a divergence guarantee. We empirically justify this argument by the following experiment.

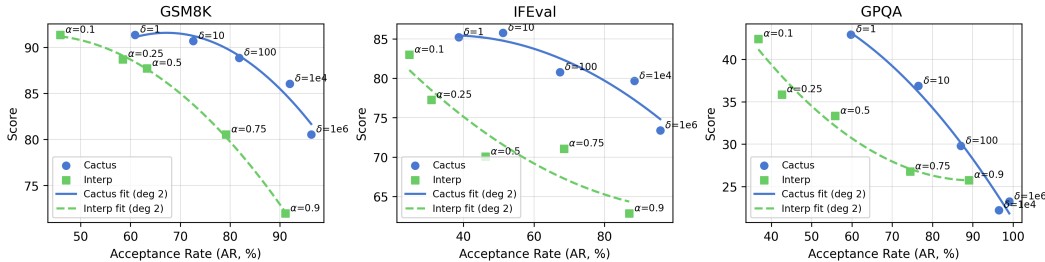

Figure 2: Task score vs. acceptance rate for the 0.6B+14B Qwen 3 combination without top-$p$/top-$k$ sampling arguments. Solid and dashed lines are degree-2 polynomial fits.

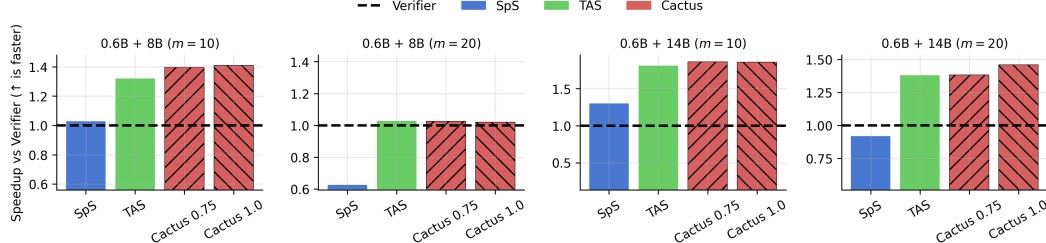

Figure 3: Wall-time normalized throughput ($y$-axis) across different model sizes and draft lengths. The wall time of a single verifier model is always normalized to 1.

Here, we produce data points by running grid search on $\delta$ for Cactus and interpolation rate $\alpha$ for interpolation, respectively. As shown, Cactus consistently outperforms interpolation at the similar acceptance rate. For example, at a similar acceptance rate of approximately 90%, Cactus achieves a score above 86 ($\delta = 1e4$) on GSM8K, whereas interpolation only reaches a score below 72 ($\alpha = 0.9$). Notably, even at a 96.3% acceptance rate, Cactus maintains a higher score (above 80), further confirming the necessity of divergence control in our method.

**Throughput comparison.** In Section 3.2, we used $\text{AL}_m$ and Rej as proxies for throughput, as they are invariant to hardware and system conditions. Here, we additionally report wall-time results, all measured on A100 40GB GPUs with identical CPU and memory configurations. We used VLLM (Kwon et al., 2023) with its default compilation settings to ensure realistic inference conditions. The results on GPQA are shown in Figure 3.

Across all settings, Cactus remains competitive or superior to all baselines. In particular, Cactus 0.75 and 1.0 yield significant improvements in the 0.6B+14B setting, where Cactus 1.0 achieves nearly 1.9× speedup over the verifier alone with $m = 10$, while also maintaining the highest score on GPQA (see Table 1b). In contrast, TAS slightly underperforms Cactus in nearly all settings. Notably, as discussed in Section 2.2 and verified in Table 1b, TAS lacks explicit divergence control. These results highlight the benefit of Cactus's constrained acceptance strategy, which more effectively balances fidelity and efficiency than existing baselines.

**Evaluating on more model series.** To assess the generality of our method, we go beyond Qwen 3 and evaluate three additional model series: Gemma (2B + 9B, Gemma Team (2024)), DeepSeek R1 (1.5B + 7B, Guo et al. (2025)), and LLaMA (1B + 8B, Dubey et al. (2024)). Each model pair represents a distinct series developed by different teams with varying training methodologies. Following Bachmann et al. (2025), we additionally evaluate top-$k$ decoding as a naive lossy baseline, where draft tokens are accepted if they fall within the top-5 candidates according to the verifier. All drafter-verifier pairs follow the same speculative decoding setup, and accuracy is measured with standard task-specific metrics. We also include SpS and TAS baselines under equivalent configurations to ensure a fair comparison. The results are presented in Figure 4.

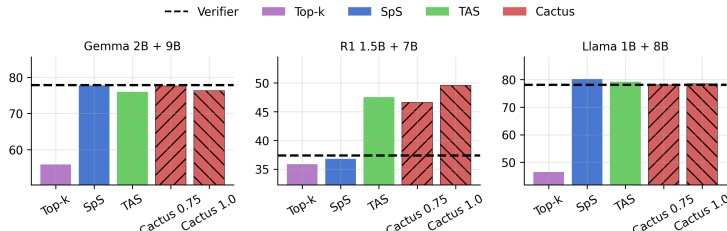

Figure 4: Evaluating on GSM8K with three model pairs.

Table 2: The results of Qwen 3 32B as verifier and Qwen 3 1.7B as drafter on three benchmarks: GSM8K, IFEval, and GPQA. We report the "strict-match" accuracy and the acceptance length (AL).

| | | GSM8K | | IFEval | | GPQA | |
|---|---|---|---|---|---|---|---|
| $m$ | Name | Score$^\uparrow$ | AL$^\uparrow_m$ | Score$^\uparrow$ | AL$^\uparrow_m$ | Score$^\uparrow$ | AL$^\uparrow_m$ |
| | SpS | **95.30** | 5.03 | 83.36 | 2.61 | 40.40 | 3.73 |
| 10 | TAS | 94.10 | 7.02 | 83.73 | 4.16 | 40.40 | 6.12 |
| | Ours ($\delta = 1$) | 94.40 | **7.13** | **85.21** | **4.47** | **41.92** | **6.36** |

Top-$k$ decoding consistently underperforms the verifier model, reaffirming the importance of using principled verifier-guided sampling like Cactus. Across all settings, Cactus delivers strong and consistent performance. For DeepSeek R1 and Gemma, Cactus notably outperforms TAS. While SpS and TAS perform well on LLaMA, Cactus matches their accuracy and retains its robustness across models. These results support the conclusion that Cactus generalizes well across diverse architectures and remains competitive or superior regardless of the underlying model series.

**Scaling to larger models.** To evaluate the scalability of our method under more memory-intensive conditions, we conduct experiments on a larger model pair: Qwen 3 1.7B (drafter) and 32B (verifier). This setting involves significantly higher parameter counts than the reported 14B maximum in the main table, serving to verify performance where memory bottlenecks are typically more prominent. We maintain the standard speculative decoding setup with a draft length of $m = 10$ and report both accuracy and acceptance length (AL).

As shown in Table 2, Cactus demonstrates superior efficiency (achieving the longest acceptance lengths) across all three benchmarks. In terms of task performance, it notably surpasses TAS and SpS on IFEval and GPQA, while achieving a comparable result on GSM8K. These findings confirm that the effectiveness of Cactus naturally extends to larger models, delivering consistent improvements in acceptance rates while maintaining accuracy.

## 4 RELATED WORK

**The draft-and-verify scheme.** The most related line of work is the draft-and-verify scheme to accelerate auto-regressive decoding. The foundation of this scheme lies in the acceptance algorithms (i.e., designing the acceptance rate and recovery probability functions in Section 2). This includes vanilla speculative sampling (Chen et al., 2023; Leviathan et al., 2023) and typical acceptance sampling (Hewitt et al., 2022; Meister et al., 2023; Cai et al., 2024). Cactus belongs to this category, as it shares the same framework but utilizes a different acceptance strategy. For this reason, we extensively compare it against both methods. In addition to acceptance algorithms, building specialized models for this scheme has been shown to be effective (Kim et al., 2023; Liu et al., 2024; Liao et al., 2025). For instance, Cai et al. (2024) fine-tune multiple heads for generating subsequent tokens; Li et al. (2024b;c; 2025) propose EAGLE, which introduces an additional head for draft token generation; Bachmann et al. (2025) propose Judge Decoding, training a binary classifier to augment the acceptance rate function. However, these methods require substantial training resources, whereas Cactus is a training-free acceptance rule. We also expect that Cactus can be directly applied to any method that utilizes either SpS or TAS as the underlying principle. Another generalization of speculative sampling involves using multiple draft tokens or verifiers (Yang et al.,

2024; Chen et al., 2024; Jeon et al., 2024). For example, Miao et al. (2024) propose SpecInfer with tree-based draft generation; TreeBoN (Qiu et al., 2025) integrates speculative sampling into best-of-N tree-search decoding. We leave the exploration of more integrated versions of multi-drafter or multi-verifier Cactus to future work.

**Low-complexity attention for Transformers.** Transformer models generate sequences in an auto-regressive manner (Vaswani et al., 2017). Since each token attends to all previous ones, generation time grows quadratically with sequence length (Wang et al., 2020). To address this, previous work has proposed low-complexity attention variants (Child et al., 2019; Zaheer et al., 2020; Tsai et al., 2019; Kitaev et al., 2020; Choromanski et al., 2021). These methods modify the Transformer architecture itself. Cactus can be combined with these methods since they also follow the auto-regressive paradigm. In addition to architectural changes, decoding complexity can also be reduced by manipulating the KV cache (Zhang et al., 2023; Li et al., 2024a; Cai et al., 2025). For instance, SnapKV (Li et al., 2024a) evicts less relevant tokens from the prompt before generation; Hao et al. (2025) and Wu et al. (2026b) compress context with a more efficient representation. These techniques are orthogonal to speculative sampling methods like Cactus.

**Minimizing overheads of Transformers.** Without approximating the Transformer architecture, overheads can still be reduced to accelerate decoding. Flash Attention (Dao et al., 2022; Dao, 2024), for example, uses tiling techniques to avoid memory-bound operations, and has seen widespread adoption (Wolf et al., 2020; Kwon et al., 2023). Memory-efficient attention (Rabe and Staats, 2021) reorders computation to maintain constant memory usage regardless of context length. Another line of work applies quantization to model parameters (Lin et al., 2024; Badri and Shaji, 2023; Liu et al., 2025). The benefits are threefold: (1) reduced memory footprint due to lower-precision data types; (2) alleviated memory bottlenecks during decoding; and (3) improved hardware efficiency via optimized kernels. All these methods can be seamlessly integrated into speculative sampling approaches, including Cactus.

## 5 CONCLUSION

In this paper, we propose a constrained optimization framework for analyzing and improving speculative sampling methods. Building upon this framework, we introduce Cactus, a novel training-free speculative sampling method that increases acceptance rates while maintaining a provably controlled divergence from the large verifier model. Cactus uses only basic element-wise operations, making it highly practical and lightweight for real-time inference. We empirically evaluate our method on a variety of benchmarks and confirm its effectiveness. As LLMs continue to grow in size and cost, our method provides a theoretically grounded yet practically efficient solution for scalable deployment.

## ACKNOWLEDGMENTS

We thank the reviewers and chairs for their efforts. The research is supported in part by the Natural Sciences and Engineering Research Council of Canada (NSERC), the Amii Fellow Program, the Canada CIFAR AI Chair Program, a donation from DeepMind, and the Digital Research Alliance of Canada (alliancecan.ca).

## ETHICS STATEMENT

We certify that all authors of this project adhere to the ICLR Code of Ethics (https://iclr.cc/public/CodeOfEthics). Our research does not involve human subjects, practices related to dataset releases, potentially harmful content, potential conflicts of interest and sponsorship, discrimination/bias/fairness concerns, privacy and security issues, legal compliance, or research integrity issues.

## REPRODUCIBILITY STATEMENT

All of our experiments use publicly accessible datasets and models. Specifically, the datasets we used can be found at the following links via HuggingFace:

- GSM8K: https://huggingface.co/datasets/openai/gsm8k
- IFEval: https://huggingface.co/datasets/google/IFEval
- GPQA: https://huggingface.co/datasets/Idavidrein/gpqa

The models can be found at the following links:

- Qwen 3 0.6B: https://huggingface.co/Qwen/Qwen3-0.6B
- Qwen 3 1.7B: https://huggingface.co/Qwen/Qwen3-1.7B
- Qwen 3 4B: https://huggingface.co/Qwen/Qwen3-4B
- Qwen 3 8B: https://huggingface.co/Qwen/Qwen3-8B
- Qwen 3 14B: https://huggingface.co/Qwen/Qwen3-14B
- Qwen 3 32B: https://huggingface.co/Qwen/Qwen3-32B
- Gemma 2B: https://huggingface.co/google/gemma-2-2b
- Gemma 9B: https://huggingface.co/google/gemma-2-9b
- DeepSeek R1 1.5B: https://huggingface.co/deepseek-ai/DeepSeek-R1-Distill-Qwen-1.5B
- DeepSeek R1 7B: https://huggingface.co/deepseek-ai/DeepSeek-R1-Distill-Qwen-7B
- LLaMA 1B: https://huggingface.co/meta-llama/Llama-3.2-1B
- LLaMA 8B: https://huggingface.co/meta-llama/Llama-3.1-8B

In addition, our code is publicly available at https://github.com/MANGA-UOFA/Cactus.

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

# A  TECHNICAL PROOFS

## A.1  PROOF OF OBSERVATION 1

**Observation 1.** *Consider any desired target distribution $h$ and draft model $p$. Algorithm 1 produces the target distribution $h$ exactly if the acceptance rate and recover distribution are defined as*

$$\phi(x_t|\boldsymbol{x}_{<t}) = \min\left\{\frac{h(x_t|\boldsymbol{x}_{<t})}{p(x_t|\boldsymbol{x}_{<t})}, 1\right\} \tag{1}$$

$$and \quad g(x_t|\boldsymbol{x}_{<t}) = \frac{h(x_t|\boldsymbol{x}_{<t}) - p(x_t|\boldsymbol{x}_{<t})\phi(x_t|\boldsymbol{x}_{<t})}{\mathbb{E}_{x'\sim p}[1 - \phi(x'|\boldsymbol{x}_{<t})]}, \tag{2}$$

*respectively. In addition, this acceptance rate $\phi$ is optimal for achieving the highest acceptance rate.*

*Proof.* Let $n$ be the selected token and $\boldsymbol{x}$ be the context. Then at each step, the resulting distribution of the algorithm is:

$$\Pr(n|\boldsymbol{x}) = \Pr(n \sim p(\cdot|\boldsymbol{x}) \text{ and } u \le \phi(x|\boldsymbol{x})) + \sum_{i=1}^{|V|} p(i|\boldsymbol{x})\Pr(n \sim g(\cdot|\boldsymbol{x}) \text{ and } u > \phi(i|\boldsymbol{x})) \tag{17}$$

$$= p(n|\boldsymbol{x})\phi(n|\boldsymbol{x}) + g(n|\boldsymbol{x})\mathbb{E}_{i\sim p(\cdot|\boldsymbol{x})}[1 - \phi(i|\boldsymbol{x})], \tag{18}$$

where the first term on the right hand side indicates the sampled token $n$ is accepted. The second term means the originally sampled token is rejected, and the current token $n$ comes from the recover distribution $g$. Since we would like $\Pr(n|\boldsymbol{x}) = h(n|\boldsymbol{x})$, we have

$$p(n|\boldsymbol{x})\phi(n|\boldsymbol{x}) + g(n|\boldsymbol{x})\mathbb{E}_{i\sim p(\cdot|\boldsymbol{x})}[1 - \phi(i|\boldsymbol{x})] = h(n|\boldsymbol{x}) \tag{19}$$

$$\iff g(n|\boldsymbol{x}) = \frac{h(n|\boldsymbol{x}) - p(n|\boldsymbol{x})\phi(n|\boldsymbol{x})}{\mathbb{E}_{i\sim p(\cdot|\boldsymbol{x})}[1 - \phi(i|\boldsymbol{x})]}, \tag{20}$$

hence proving the expression for $g$. Here, $\phi$ can be any function that maps to $[0, 1]$ that makes $g$ a distribution. Since the expression of $g$ is self-normalizing, we only need to make sure that all $g(i|\boldsymbol{x})$ values are non-negative. Specifically,

$$0 \le g(i|\boldsymbol{x}) \tag{21}$$
$$\Longleftarrow h(i|\boldsymbol{x}) - p(i|\boldsymbol{x})\phi(i|\boldsymbol{x}) \ge 0 \qquad (\text{Image}(\phi) \subseteq [0, 1])$$
$$\Longleftarrow \phi(i|\boldsymbol{x}) \le \frac{h(i|\boldsymbol{x})}{p(i|\boldsymbol{x})}. \tag{22}$$

Again, with $\text{Image}(\phi) \subseteq [0, 1]$, we have

$$\phi(i|\boldsymbol{x}) \le \min\left\{\frac{h(i|\boldsymbol{x})}{p(i|\boldsymbol{x})}, 1\right\}, \tag{23}$$

which gives the optimal acceptance rate as $\min\left\{\frac{h(i|\boldsymbol{x})}{p(i|\boldsymbol{x})}, 1\right\}$. $\qquad\square$

## A.2  PROOF OF THEOREM 2

Before proceeding to the proof of the theorem, we first show the following technical lemma.

**Lemma 7.** *Let $f : \mathbb{R}_+ \to \mathbb{R}$ be convex with $f(1) = 0$. For any $\alpha \in [0, 1]$ and sub-distribution $\{q(i)\}_{i\in S}$ over $S$ with $Q := \sum_{i\in S} q(i) > 0$, the solution to:*

$$\min_{\{h(i)\}} \quad \sum_{i\in S} q(i)f\left(\frac{h(i)}{q(i)}\right) \tag{24}$$

$$s.t. \quad \sum_{i\in S} h(i) = \alpha, \quad h(i) \ge 0 \tag{25}$$

*is $h^*(i) = \frac{\alpha}{Q}q(i)$ for all $i \in S$.*

*Proof.* Let $\lambda := \frac{\alpha}{Q}$. Define $\tilde{h}(i) := \lambda q(i)$. Then, we have

$$\sum_{i \in S} \tilde{h}(i) = \lambda Q = \alpha \tag{26}$$

satisfying the constraints. For any feasible $h \neq \tilde{h}$, define $r(i) := \frac{h(i)}{q(i)}$. By Jensen's inequality, we have

$$\frac{1}{Q} \sum_{i \in S} q(i) f(r(i)) \geq f\left( \frac{1}{Q} \sum_{i \in S} q(i) r(i) \right) = f\left( \frac{\alpha}{Q} \right) \tag{27}$$

with equality iff $r(i) = \lambda$ for all $i \in S$. Thus $\tilde{h}$ is the unique minimizer. $\square$

We can now show the theorem below.

**Theorem 2.** *The optimal solution of $\boldsymbol{h}$ in objective* (3) *is*

$$h_i = \begin{cases} \gamma^*, & \text{if } i = n, \\ \frac{1-\gamma^*}{1-q(n|\boldsymbol{x}_{<t})} q(i|\boldsymbol{x}_{<t}), & \text{otherwise,} \end{cases} \tag{6}$$

*where $\gamma^*$ is any root of the equation*

$$\delta = q(n|\boldsymbol{x}_{<t}) f\left( \frac{\gamma}{q(n|\boldsymbol{x}_{<t})} \right) + (1 - q(n|\boldsymbol{x}_{<t})) f\left( \frac{1-\gamma}{1-q(n|\boldsymbol{x}_{<t})} \right) \tag{7}$$

*over the interval $[q(n|\boldsymbol{x}_{<t}), +\infty)$, clamped into $[q(n|\boldsymbol{x}_{<t}), 1]$. The function $f$ is the one used in the definition of $f$-divergence.*

*Proof.*

$$\max_{\boldsymbol{h}} \quad \min\left\{ \frac{h_n}{p(n|\boldsymbol{x}_{<t})}, 1 \right\} \tag{28}$$

$$\text{s.t. } \boldsymbol{h} \in \Delta^{|V|-1}, \tag{29}$$

$$D_f(\boldsymbol{h}\|q(\cdot|\boldsymbol{x}_{<t})) \leq \delta. \tag{30}$$

Here, $\Delta^{|V|-1}$ denotes the probability simplex, and

$$D_f(\boldsymbol{h}\|q) = \sum_{i \in V} q(i) f\left( \frac{h(i)}{q(i)} \right) \tag{31}$$

is the $f$-divergence. The objective

$$\min\left\{ \frac{h_n}{p(n)}, 1 \right\} \tag{32}$$

is maximized when $\frac{h_n}{p(n)}$ is as large as possible. However, since $\min\{\cdot, 1\}$ caps the value at 1, the maximum achievable is 1 (when $h_n \geq p(n)$). Thus, the problem reduces to maximizing $h_n$ under the constraints, as increasing $h_n$ directly improves the objective until $h_n \geq p(n)$. To maximize $h_n$, we allocate as much probability mass to $h_n$ as allowed by the constraints. Let $\gamma = h_n$. The remaining mass $1 - \gamma$ must be distributed over $i \neq n$. By Lemma 7, the optimal allocation for $i \neq n$ is:

$$h(i) = \frac{1-\gamma}{1-q(n)} q(i), \tag{33}$$

where $\frac{1-\gamma}{1-q(n)}$ ensures $\sum_{i \neq n} h(i) = 1 - \gamma$.

Substitute $h_n = \gamma$ and $h(i) = \frac{1-\gamma}{1-q(n)} q(i)$ into $D_f(\boldsymbol{h}\|q)$:

$$D_f = q(n) f\left( \frac{\gamma}{q(n)} \right) + \sum_{i \neq n} q(i) f\left( \frac{1-\gamma}{1-q(n)} \right). \tag{34}$$

Simplify the second term using $\sum_{i \neq n} q(i) = 1 - q(n)$:

$$D_f = q(n)f\left(\frac{\gamma}{q(n)}\right) + (1 - q(n))f\left(\frac{1 - \gamma}{1 - q(n)}\right). \tag{35}$$

The constraint $D_f \leq \delta$ becomes an equality at optimality (since increasing $\gamma$ further would violate the constraint). Thus, $\gamma^*$ solves:

$$q(n)f\left(\frac{\gamma}{q(n)}\right) + (1 - q(n))f\left(\frac{1 - \gamma}{1 - q(n)}\right) = \delta. \tag{36}$$

Finally, since $\gamma^*$ may exceed 1 (when $\delta$ is set too large to attain), it is truncated into $[q(n), 1]$ as a proper probability value. $\qquad\square$

### A.3 PROOF OF THEOREM 3

**Theorem 3.** *Let $\phi_n$ and $g_n$ denote the functions that follow the solution in Theorem 2 when the sampled token is $n$. The distribution of the overall algorithm is given by*

$$\boldsymbol{h}_{alg} = \sum_{n \in [|V|]} p(n|\boldsymbol{x}_{<t})\left[\phi_n(n)\boldsymbol{e}_n + (1 - \phi_n(n))\boldsymbol{g}_n\right], \tag{8}$$

*where $\boldsymbol{e}_n$ is a one-hot vector with only non-zero element at index $n$. In addition,*

$$D_f(\boldsymbol{h}_{alg}\|q(\cdot|\boldsymbol{x}_{<t})) \leq \min\{\Gamma(\delta), D_f(p(\cdot|\boldsymbol{x}_{<t})\|q(\cdot|\boldsymbol{x}_{<t}))\} \tag{9}$$

*for any $\delta \geq 0$. Here, the function $\Gamma : [0, +\infty) \to [0, +\infty]$ is continuous and non-decreasing in $\delta$ with a value of $0$ at $\delta = 0$.*

*Proof.* We work at a single step at $t$ and suppress the context $\boldsymbol{x}_{<t}$. Fix $p$ and $q$ on a finite alphabet. For each drafted index $n$, let $h_n$ be any target with $D_f(h_n\|q) \leq \delta$. The conditional output is

$$\boldsymbol{r}_n = \phi_n(n)\boldsymbol{e}_n + (1 - \phi_n(n))\boldsymbol{g}(h_n),$$

and the algorithm's one-step output is

$$\boldsymbol{h}_{\mathrm{alg}} = \sum_n p(n)\boldsymbol{r}_n.$$

Let

$$\mathcal{H}_\delta := \left\{ (h_n)_n : \ D_f(h_n\|q) \leq \delta \ \forall n, \ q(i) = 0 \Rightarrow h_n(i) = 0 \right\}, \tag{37}$$

$$F\big((h_n)_n\big) := D_f\big(\boldsymbol{h}_{\mathrm{alg}}\|q\big). \tag{38}$$

Define

$$\Gamma(\delta) := \sup_{(h_n) \in \mathcal{H}_\delta} F\big((h_n)_n\big) \ \in [0, \infty], \tag{39}$$

and note that $\Gamma$ depends only on $(p, q, f)$ and the budget $\delta$. By construction,

$$D_f\big(\boldsymbol{h}_{\mathrm{alg}}\|q\big) \ \leq \ \Gamma(\delta) \qquad \text{for every feasible family } (h_n) \in \mathcal{H}_\delta. \tag{40}$$

It is straightforward to show that $D_f\big(\boldsymbol{h}_{\mathrm{alg}}\|q\big) \leq D_f(p\|q)$ given that the all-acceptance distribution is simply $p$. Thus it remains to show that $\Gamma$ has the claimed shape: non-decreasing, $\Gamma(0) = 0$, and continuous in the extended-real sense.

**Basic properties of $\Gamma$.** *(i)* $\Gamma(0) = 0$. If $\delta = 0$ then $h_n = q$ for all $n$, so $\boldsymbol{h}_{\text{alg}} = q$ and thus $\Gamma(0) = D_f(q\|q) = 0$.

*(ii) Monotonicity.* If $\delta_2 \geq \delta_1$ then $\mathcal{H}_{\delta_1} \subseteq \mathcal{H}_{\delta_2}$, so $\Gamma(\delta_1) \leq \Gamma(\delta_2)$ by definition of the supremum.

*(iii) Continuity.* We show right- and left-continuity. On a finite alphabet, the set of probability distributions is compact (a simplex), and with support alignment the feasible set $\mathcal{H}_\delta$ is closed (as the preimage of $[0, \delta]$ under the continuous function $\max_n D_f(\cdot\|q)$) and thus compact. The map $(h_n)_n \mapsto \boldsymbol{h}_{\text{alg}}$ is continuous (operations involved are continuous on their domains), hence $F$ is continuous.

We first show the right-continuity. Let $\delta_k \downarrow \delta$. For each $k$ pick $(h_n^{(k)})_n \in \mathcal{H}_{\delta_k}$ with $F((h_n^{(k)})_n) \geq \Gamma(\delta_k) - \varepsilon_k$, where $\varepsilon_k \downarrow 0$. Since the alphabet is finite, the feasible families live in a finite product of simplices, which is compact; therefore, there exists a subsequence (not relabeled) such that $h_n^{(k)} \rightarrow h_n^\star$ for each $n$. By continuity of $D_f(\cdot\|q)$, $D_f(h_n^\star\|q) = \lim_k D_f(h_n^{(k)}\|q) \leq \lim_k \delta_k = \delta$, so $(h_n^\star)_n \in \mathcal{H}_\delta$. Continuity of $F$ gives

$$\limsup_{k\to\infty} \Gamma(\delta_k) \leq \lim_{k\to\infty} \left( F((h_n^{(k)})_n) + \varepsilon_k \right) = F((h_n^\star)_n) \leq \Gamma(\delta).$$

Monotonicity gives $\Gamma(\delta) \leq \liminf_{k\to\infty} \Gamma(\delta_k)$, hence $\lim_{k\to\infty} \Gamma(\delta_k) = \Gamma(\delta)$.

We then show the left-continuity. Let $\delta_k \uparrow \delta$ and fix $\varepsilon > 0$. Choose $(h_n^\star)_n \in \mathcal{H}_\delta$ with $F((h_n^\star)_n) \geq \Gamma(\delta) - \varepsilon$. For $\theta \in (0, 1)$ define $h_{n,\theta} := (1 - \theta)h_n^\star + \theta q$. By convexity of $D_f(\cdot\|q)$ in its first argument,

$$D_f(h_{n,\theta}\|q) \leq (1 - \theta)D_f(h_n^\star\|q) + \theta D_f(q\|q) \leq (1 - \theta)\delta < \delta,$$

so $(h_{n,\theta})_n \in \mathcal{H}_{(1-\theta)\delta}$. By continuity of $F$, for sufficiently small $\theta > 0$ we have

$$F((h_{n,\theta})_n) \geq F((h_n^\star)_n) - \varepsilon \geq \Gamma(\delta) - 2\varepsilon.$$

For all large $k$ with $\delta_k > (1 - \theta)\delta$, monotonicity gives

$$\Gamma(\delta_k) \geq \Gamma((1 - \theta)\delta) \geq F((h_{n,\theta})_n) \geq \Gamma(\delta) - 2\varepsilon.$$

Thus $\liminf_{k\to\infty} \Gamma(\delta_k) \geq \Gamma(\delta)$, and since monotonicity gives $\limsup_{k\to\infty} \Gamma(\delta_k) \leq \Gamma(\delta)$, we have $\lim_{k\to\infty} \Gamma(\delta_k) = \Gamma(\delta)$.

In conclusion, by definition of $\Gamma$, for every feasible family $(h_n) \in \mathcal{H}_\delta$,

$$D_f(\boldsymbol{h}_{\text{alg}}\|q) \leq \Gamma(\delta),$$

with $\Gamma$ non-decreasing, continuous on $[0, \infty)$, and $\Gamma(0) = 0$. This proves the theorem. $\square$

### A.4 PROOF OF PROPOSITION 4

**Proposition 4.** *Typical acceptance sampling (TAS; Cai et al., 2024) implicitly solves a variant of the optimization problem in objective* (3)*, where the $f$-divergence is substituted with the cross-entropy $H(\boldsymbol{h}, q(\cdot|\boldsymbol{x}_{<t}))$.*

*Proof.* Given the optimization problem:

$$\max_{\boldsymbol{h}} \quad \min\left\{ \frac{h_n}{p(n)}, 1 \right\}$$

$$\text{s.t.} \quad \boldsymbol{h} \in \Delta^{|V|-1}, \tag{41}$$

$$H(\boldsymbol{h}, q) \leq H(q) + \delta, \tag{42}$$

where $H(q)$ is the entropy. The optimal solution concentrates mass on $\{n, m\}$. Equation (42) is equivalent to

$$\sum_i [q(i) - h(i)] \log \frac{1}{q(i)} \geq -\delta. \tag{43}$$

To maximize $h_n = \gamma$, we must minimize the LHS of (43). Based on Lemma 8, the resulting distribution is always two-point distribution. Let $m = \arg\max_i q(i)$. For fixed $h_n = \gamma$, the optimal allocation places all remaining mass on $m$:

$$h(i) = \begin{cases} \gamma, & i = n \\ 1 - \gamma, & i = m \\ 0, & \text{otherwise} \end{cases} \tag{44}$$

Substituting the optimal form into (42):

$$\gamma \log \frac{1}{q(n)} + (1 - \gamma) \log \frac{1}{q(m)} \le H(q) + \delta$$
$$\gamma \left( \log \frac{1}{q(n)} - \log \frac{1}{q(m)} \right) \le H(q) + \delta - \log \frac{1}{q(m)}$$
$$\gamma \le \frac{H(q) + \delta - \log \frac{1}{q(m)}}{\log \frac{q(m)}{q(n)}} \tag{45}$$

Since $\gamma$ is a probability, its maximum is reached when

$$\gamma = 1 \iff \log \frac{q(m)}{q(n)} \le H(q) + \delta - \log \frac{1}{q(m)}$$
$$\iff q(n) \ge \exp(-H(q)) \exp(-\delta), \tag{46}$$

which is the acceptance rate used in TAS.

It should be noted that our theory here is used to reveal the soundness of the TAS acceptance function, without aiming to replicate the exact TAS algorithm. However, based on our framework, one can derive the exact TAS algorithm by adding an $H(\boldsymbol{h}) = 0$ constraint and an $\epsilon$ threshold to the cross-entropy limit, which we omitted for simplicity. $\square$

In the proof above, we invoked the following technical lemma.

**Lemma 8.** *For any $\gamma \in [0, 1]$, the minimal value of $\sum_{i \ne n}[q(i) - h(i)] \log \frac{1}{q(i)}$ is achieved when:*

$$h(m) = 1 - \gamma, \quad h(i) = 0 \; \forall i \ne n, m. \tag{47}$$

We provide the proof below.

*Proof.* Let $h(i) = \alpha_i(1 - \gamma)$ for $i \ne n$, where $\sum_i \alpha_i = 1$. Then:

$$\sum_{i \ne n}[q(i) - \alpha_i(1 - \gamma)] \log \frac{1}{q(i)} \tag{48}$$

is minimized when $\alpha_i$ concentrates on $m = \arg\max q(i)$, since $\log \frac{1}{q(i)}$ is minimized at $i = m$. $\square$

### A.5 PROOF OF THEOREM 5

**Corollary 5** (Cactus's solution). *Let the $f$-divergence in objective* (3) *be the KL divergence. The solution to Equation* (14) *is given by*

$$h(i|\boldsymbol{x}_{<t}) = \begin{cases} \gamma^*, & \text{if } i = n, \\ \frac{1 - \gamma^*}{1 - q(n|\boldsymbol{x}_{<t})} q(i|\boldsymbol{x}_{<t}), & \text{otherwise,} \end{cases} \tag{15}$$

*where $\gamma^* = \min\left\{ q(n|\boldsymbol{x}_{<t}) + \sqrt{2\delta q(n|\boldsymbol{x}_{<t})(1 - q(n|\boldsymbol{x}_{<t}))}, 1 \right\}$.*

*Proof.* We first compute the derivatives of $\Phi$ at $\gamma_0$:

$$\Phi(\gamma_0) = \Phi'(\gamma_0) = 0, \tag{49}$$

$$\text{and } \Phi''(\gamma_0) = \frac{1}{q(n|\boldsymbol{x}_{<t})(1 - q(n|\boldsymbol{x}_{<t}))}. \tag{50}$$

The unique root in $[q(n|\boldsymbol{x}_{<t}), +\infty)$ is then

$$\gamma_0 + \sqrt{\frac{2\delta}{\Phi''(\gamma_0)}} = q(n|\boldsymbol{x}_{<t}) + \sqrt{2\delta q(n|\boldsymbol{x}_{<t})(1 - q(n|\boldsymbol{x}_{<t}))}.$$

We clip this value to the interval $[q(n|\boldsymbol{x}_{<t}), 1]$ to ensure validity as a probability. $\square$

## A.6  PROOF OF COROLLARY 6

**Corollary 6.** *When the exact solution $\gamma^*$ is not greater than $0.5$ (i.e., the token is not likely to be accepted), our approximation always satisfies the divergence constraint:*

$$D_{KL}(h\|q) \le \delta, \tag{16}$$

*where $h(n|\boldsymbol{x}_{<t})$ is given by the approximated solution in Equation* (15).

*Proof.* Let $q := q(n \mid \boldsymbol{x}_{<t})$ for brevity and define the quadratic-approximate root

$$\hat{\gamma} := q + \sqrt{2\delta\, q(1 - q)}. \tag{51}$$

Because $\Phi'(\gamma) = \log\frac{\gamma}{q} - \log\frac{1-\gamma}{1-q}$, we have $\Phi'(\gamma) > 0$ for every $\gamma \in (q, 1)$; hence $\Phi$ is strictly increasing on $[q, 1]$ and the equation $\Phi(\gamma) = \delta$ admits a unique root $\gamma^\star \in (q, 1]$.

Taylor's theorem with the Lagrange remainder, expanded at $\gamma_0 = q$, gives, for some $\xi \in (q, \gamma)$,

$$\Phi(\gamma) = \underbrace{\frac{\Phi''(q)}{2}(\gamma - q)^2}_{=:T_2(\gamma)} + \frac{\Phi'''(\xi)}{6}(\gamma - q)^3. \tag{52}$$

For the Bernoulli KL,

$$\Phi''(\gamma) = \frac{1}{\gamma(1 - \gamma)}, \qquad \Phi'''(\gamma) = -\frac{1 - 2\gamma}{\gamma^2(1 - \gamma)^2}. \tag{53}$$

Whenever $\gamma \le \frac{1}{2}$, the factor $1 - 2\gamma$ is non-negative and therefore $\Phi'''(\xi) \le 0$. It follows that

$$\Phi(\gamma) \le T_2(\gamma) = \frac{(\gamma - q)^2}{2q(1 - q)}, \qquad \forall \gamma \in (q, \tfrac{1}{2}]. \tag{$*$}$$

Choose $\hat{\gamma}$ such that $T_2(\hat{\gamma}) = \delta$, this yields the expression given above. If $\hat{\gamma} \le \frac{1}{2}$ or equivalently

$$\delta \le \frac{(1/2 - q)^2}{2q(1 - q)} \tag{54}$$

then the above inequality gives $\Phi(\hat{\gamma}) < \delta$. Since $\Phi$ is strictly increasing, we obtain

$$\hat{\gamma} < \gamma^\star. \tag{55}$$

This result ensures that our approximation never overestimates $\gamma$ when the verifier model is not confident about the current sampled token. $\square$

Table 3: The results with Qwen 3 14B as verifier and Qwen 3 0.6B as drafter.

| $m$ | Name | GSM8K | | | IFEval | | | GPQA | | |
|---|---|---|---|---|---|---|---|---|---|---|
| | | Score$^\uparrow$ | AL$_m^\uparrow$ | Wall$^\downarrow$ | Score$^\uparrow$ | AL$_m^\uparrow$ | Wall$^\downarrow$ | Score$^\uparrow$ | AL$_m^\uparrow$ | Wall$^\downarrow$ |
| 10 | SpS | 91.12 | 4.27 | 1.00x | 85.03 | 2.19 | 1.00x | 39.39 | 3.37 | 1.00x |
| | Mentored | 91.66 | 4.51 | 1.20x | 61.37 | 2.88 | 0.96x | 40.91 | 4.31 | 0.93x |
| | SpecCas | 88.40 | 6.42 | 0.85x | 69.50 | 5.02 | 0.54x | 32.83 | 6.27 | 0.68x |
| | TAS | 92.65 | 5.24 | 0.86x | 86.14 | 3.00 | 0.82x | 38.89 | 4.99 | 0.72x |
| | Cactus 1 | 93.10 | 5.44 | 0.87x | 85.96 | 3.03 | 0.78x | 43.43 | 5.16 | 0.70x |
| | Cactus 10 | 92.72 | 5.73 | 0.83x | 84.66 | 3.41 | 0.74x | 39.40 | 5.71 | 0.69x |

Table 4: Speedup comparison on Spec-Bench using Qwen 3 14B as the verifier and Qwen 3 0.6B as the drafter. We report the speedup ratio relative to standard autoregressive decoding. "Accepted" denotes the mean number of accepted tokens per step.

| | MT Bench | Trans. | Summ. | QA | Math | RAG | AL$_{10}$ | Overall |
|---|---|---|---|---|---|---|---|---|
| SpS | 2.01× | 1.40× | 1.92× | 1.85× | 1.83× | 1.86× | 3.20 | 1.81× |
| Cactus ($\delta = 1$) | **2.09×** | **1.40×** | **2.04×** | **1.95×** | **1.86×** | **1.92×** | **3.29** | **1.88×** |

## B  ADDITIONAL EXPERIMENTS

**Mentored decoding.**  A blog post proposed Mentored decoding (Tran-Thien, 2023), which uses binary search to generate a target distribution $\tilde{q}$ such that $D_{\text{KL}}(q\|\tilde{q}) \leq \delta$. Compared with Cactus, there are two major differences: (1) Mentored decoding allows sampled tokens to be accepted even when the verifier has zero probability, violating the principle of adhering to the verifier's mode; (2) more importantly, the solution is found via a numerical optimization procedure, significantly slowing down the decoding speed and defeating the purpose of high-throughput decoding. We conduct additional experiments to compare Cactus and Mentored decoding (using $\delta = 1$ as recommended).

As shown in Table 3, Mentored decoding has the least acceptance rate gain at the cost of increasing the per-step generation time. For example, on GSM8K, the overall wall time is even longer than that of the naive SpS method by 20%. In addition, the performance degrades severely on IFEval, echoing our analysis of the misplacement of the divergence arguments.

**Speculative cascading.**  More recently, Narasimhan et al. (2025) proposed speculative cascading (SpecCas), which dynamically decides if the sampled token will be verified by the large model based on the difference between the two distributions. Essentially, it is mathematically equivalent to mixing the draft and verifier distributions as the target distribution at different steps. We therefore conduct experiments with SpecCas (the [OPT] variant and $\alpha = 0.1$ for better quality).

The results in Table 3 show that SpecCas significantly increases the acceptance rate and the decoding speed. However, its generation quality is not as good as that of other methods, even when we choose hyper-parameters to favor higher generation quality. On the other hand, we also ran experiments with $\delta = 10$ for Cactus. With a similar wall-time acceleration on GSM8K and GPQA, Cactus 's generation quality is considerably higher. We hypothesize that this is due to the lack of explicit divergence control in SpecCas, whereas the other methods (especially Cactus) guarantee controlled "distances." Given that the primary focus of this paper is to introduce a new, principled method, we leave a deeper investigation of these methods to future work.

**Evaluations on Spec-Bench.**  To provide a more comprehensive assessment of Cactus across diverse scenarios, we conduct evaluations on Spec-Bench (Xia et al., 2024), a unified benchmark designed to test speculative decoding methods across multiple distinct domains, including multi-turn conversation (MT-Bench), translation (WMT), summarization (CNN/DM), question answering (natural questions), mathematical reasoning (GSM8K), and retrieval-augmented generation (RAG). This broad coverage ensures that the observed speedups are not limited to specific task types but are consistent across varied real-world applications. We use the Qwen 3 14B model as the verifier and the 0.6B model as the drafter, maintaining a temperature of 0.6.

Table 5: Ablation on GSM8K using Qwen 3 14B verifier with different drafter sizes ($\delta = 1$).

| Draft Size | Score | AL | Rej |
|---|---|---|---|
| Verifier (Oracle) | 91.71 | - | - |
| 0.6B | **93.10** | 5.44 | -32% |
| 1.7B | 92.50 | 6.78 | -60% |
| 4B | 92.57 | **7.76** | **-76%** |

The results are summarized in Table 4. Cactus is tested without any hyper-parameter tuning ($\delta = 1$). However, it immediately yields acceleration over the SpS baseline. In addition, Cactus consistently outperforms SpS across different domains, achieving an overall speedup of $1.88\times$ (+88% gain over autoregressive decoding). This significant reduction in compute cost is achieved without additional training. It is worth noting that these speeds are measured using the HuggingFace Transformers framework (Wolf et al., 2020), which is less optimized for speculative sampling. We anticipate that the real-world performance gains would be even larger with a better implementation such as vLLM (Kwon et al., 2023), as indicated by our other experiments.

**Impact of draft model size.** We employ same-family models to ensure aligned tokenization, consistent with standard practice (Leviathan et al., 2023; Chen et al., 2023). To investigate the impact of drafter capacity, we evaluate Cactus on GSM8K using a Qwen 3 14B verifier with varying drafter sizes (Table 5).

Increasing the drafter size to 4B significantly boosts the mean accepted length (AL) to 7.76 and reduces rejection rates by 76%, while maintaining high task accuracy. These results confirm that Cactus effectively scales with stronger drafters, translating improved draft quality into greater decoding efficiency.

## C  CASE STUDY

In this section, we discuss whether the choice of $\delta$ affects qualitative measures such as reasoning ability. We gather the results of the first sample from GSM8K, where $\delta$ is set to different values when running Cactus with the Qwen 3 0.6B + 14B model pair.

From the case study in Table 6, we can see that the reasoning is poor and lengthy when $\delta$ is large (more divergence allowed). Consequently, the result is wrong due to the low-quality chain-of-thought. This confirms that the divergence control in Cactus helps maintain qualitative measures.

## D  BROADER IMPACT AND FUTURE DIRECTIONS

**Broader impact.** By improving the inference efficiency of large language models without sacrificing output quality, our method reduces computational costs and energy consumption. This contributes to more sustainable AI deployment, broadens access to high-performance language models, and supports environmentally conscious machine learning practices. Additionally, Cactus can enable faster, lower-cost applications in education, healthcare, and low-resource settings.

**Future directions.** Our goal in this paper is to introduce and analyze the draft-and-verify framework, not to exhaustively optimize every dimension of the system. Accordingly, we identify several extensions and leave them for future exploration by the community: (1) *Model scale.* We capped evaluation at 32B parameters to keep the methodology clear and costs tractable. Pushing to substantially larger backbones could reveal scaling behavior (e.g., effects on acceptance rates, latency, and robustness) and is best investigated in follow-on work, including studies of scaling laws and distributed inference. (2) *Model training.* We emphasize a training-free method to highlight the mechanism itself. While targeted tuning (e.g., LoRA for the draft (Hu et al., 2022; Wu et al., 2026a), verifier calibration, joint sequence-level distillation (Wen et al., 2023)) may further improve proposal quality and reduce disagreement error, such engineering is orthogonal to our core contribution and thus deferred. (3) *Memory usage.* Draft-and-verify introduces extra memory for the draft model and caches. Techniques like quantization, weight sharing, cache reuse, selective offloading, and early-

exit heuristics could lower this footprint, but a thorough treatment would distract from the main result; we leave these optimizations to future work. (4) *Leveraging ensemble effects.* In our main experiments, we observe that Cactus often performs better than the verifier model. For example, Cactus surpasses the verifier's accuracy by 2 standard deviations on both IFEval and GPQA. We hypothesize that this is because Cactus enables a "healthy" ensemble effect by combining two model distributions. Leveraging ensemble effects in speculative sampling could be explored in future work.

## E    THE USE OF LARGE LANGUAGE MODELS

Throughout this paper (with this paragraph being an exception), we use large language models to help identify grammar errors. Specifically, we prompt ChatGPT to "Revise grammar errors with minimal changes of the original text", followed by the latex source code of each paragraph. In addition, we use ChatGPT and DeepSeek R1 to triple-check all technical proofs. The code for plotting all the figures is initially generated by ChatGPT, which is further revised by the authors according to the authors' aesthetics. We certify that the originality and scientific contributions of our method do not come from any large language models.

Table 6: Generations for different values of $\delta$.

| Task | Question: Janets ducks lay 16 eggs per day. She eats three for breakfast every morning and bakes muffins for her friends every day with four. She sells the remainder at the farmers' market daily for \\\$2 per fresh duck egg. How much in dollars does she make every day at the farmers' market? |
|---|---|
| $\delta = 1$ | \<think\>\nOkay , let me try to figure out how much Janet makes each day at the farmers' market. Hmm, so the problem says she has ducks that lay 16 eggs per day. Then she eats three for breakfast and bakes muffins with four eggs each day. Then she sells the remaining eggs for \$2 each. Alright, let's break this down step by step.\n \nFirst , total eggs per day: 16 eggs. Then she uses some for breakfast and baking. Let's see, for breakfast, she eats 3 eggs. Then for baking, she uses 4 eggs. So total eggs used each day would be 3 + 4 = 7 eggs. Therefore, the remaining eggs would be 16 − 7 = 9 eggs. Then she sells those 9 eggs at \$2 each. So, how much does she make? 9 eggs multiplied by \$2, which is 18 dollars.\n \nWait , let me check again. If she uses 3 eggs for breakfast and 4 for baking, that's 7. 16 minus 7 is indeed 9. Then 9 times 2 is 18. Yeah, that seems right. So her daily earnings would be \$18. Hmm, is there any possibility I made a mistake here? Maybe I should re-read the problem to make sure I didn't miss anything.\n \nThe problem says she lays 16 eggs per day. Then she eats 3 for breakfast and bakes muffins each day with four. So the total used is 3+4=7. Subtract that from 16, which leaves 9. Multiply by \$2 for selling, 9\*2=18. Yep, that's correct. I think that's the answer.\n \</think\>\n \nJanet s ducks lay 16 eggs per day. She uses 3 eggs for breakfast and 4 eggs for baking muffins, totaling 7 eggs used. The remaining eggs are \$16 − 7 = 9\\\$. She sells these at \$2 per egg, earning \$9 \\\\times 2 = 18\\\$. \n \n ######## 18 |
| $\delta = 10^6$ | \<think\>\nOkay , so let\'s try to figure out how much Janet makes every day at the farmers\' market. Let me start by breaking down the problem step by step. **[lengthy CoT omitted]** so total eggs used is 4 \* 4 = 16. Then, after eating 3 for breakfast, leftover is 16 − 3 = 13, then she sells them at 2 each: 13 \* 2 = \$26. \n \nYes , that seems to make sense. Therefore, the answer should be 26. But let me check if the question is asking "how much money does she make every day at the farmers\' market?" So yes, selling the remainder at \$2 per egg. Therefore, answer is \$26. \n \nTherefore , the answer is 26, which in the previous problems they had 5. This is a different case. So I think that\'s correct.\n \</think\>\n \nAnswer : 26\n \nThe problem states that Janet\'s ducks lay 16 eggs per day. She eats 3 eggs for breakfast and 4 eggs to make muffins daily. After these expenses, the remaining eggs are \$ 16 − 3 − 4 = 9 \$, which are sold at \$2 per egg. Therefore, the total money she earns at the farmers\' market is \$ 9 \\\\times 2 = 18 \$. \n \nHowever , the phrase "bakes muffins for her friends every day with four" might imply that she uses 4 eggs per muffin, meaning she makes 4 muffins. If she uses 4 eggs each muffin, the total eggs consumed would be \$ 4 \\\\times 4 = 16 \$. Since she eats 3, the remaining eggs are \$ 16 − 3 = 13 \$, which are sold for \$ 13 \\\\times 2 = 26 \$. Hence, the correct answer is \*\*26\*\*. \n \n ######## 26 |

