# OpenReview forum: "Cactus: Accelerating Auto-Regressive Decoding with Constrained Acceptance Speculative Sampling"
_ICLR.cc/2026/Conference — ICLR 2026 Poster_

### Official Review · Reviewer_PuX9 · 2025-10-30

**Soundness:** 3
**Presentation:** 3
**Contribution:** 3
**Rating:** 6
**Confidence:** 5

**Summary:**

This paper introduces Cactus, a constrained acceptance speculative sampling method aimed at accelerating auto-regressive decoding in large language models (LLMs) while preserving output distributional fidelity. The work frames speculative sampling as a constrained optimization problem balancing acceptance rate and divergence from a verifier model. Theoretical analysis yields a principled solution for constrained divergence (using KL divergence as a metric), and the authors propose a practical algorithm that maximizes accepted tokens per verification step. Empirical evaluation on multiple benchmarks (GSM8K, IFEval, and GPQA), using various pairs of drafting and verifying models, demonstrates improved throughput and competitive or superior accuracy versus classical speculative sampling and typical acceptance sampling approaches.

**Strengths:**

1. Formalization and Theoretical Rigor: The paper provides a thorough formalization of speculative sampling as a constrained optimization problem, with mathematically sound derivations (see Equations in Section 2 and supporting proofs in Appendix A). The use of $f$-divergences, primarily KL, is well-motivated and leads to explicit solutions for the acceptance/recovery functions.

2. Principled Design: Compared to prior heuristics like typical acceptance sampling (TAS), Cactus mathematically guarantees bounded divergence from the verifier distribution. The method is grounded in strong theoretical justifications (see Theorem 3, Corollaries 6 and 7, and explicit treatment of the divergence-acceptance trade-off).

3. Empirical Breadth and Depth: The experiments include a wide variety of benchmarks, LLM architectures, and model sizes. Table 1 provides detailed results across tasks and models, showing Cactus consistently offers higher acceptance rates (e.g., $\mathrm{AL}_{m}$ and Rej columns), often without sacrificing accuracy compared to SpS and TAS. Additional evaluations address wall-time throughput (Figure 3), cross-architecture robustness (Figure 4), and trade-offs with alternative approaches (interpolation and SpecCas).

4. Clarity and Transparency: Mathematical notation is mostly clear, with comprehensive appendices containing proofs and derivations. The design of the method is explained with both equations and pseudocode (Algorithm 1), and decision points are clearly highlighted.

5. Practical Impact: The method is training-free, requires only elementwise operations on output probabilities, and can be deployed to accelerate LLM inference in real-world conditions.
6. Figure/Table Engagement: Key quantitative claims are backed up by explicit references to figures and tables. For example, Table 1a and 1b directly show improvement of Cactus in both acceptance rate and quality, while Figure 1 graphically summarizes the accuracy-acceptance tradeoff across diverse settings.

**Weaknesses:**

1. Empirical Baseline Coverage and Ablations: While the paper includes SpS, TAS, Top-k, Mentored decoding, and SpecCas as baselines, it does not systematically evaluate against very recent tree-based decoding advancements (such as recursive or blockwise candidate strategies). This limits direct benchmarking of Cactus's relative value in the face of alternative parallel or hierarchical sampling frameworks. The omission can be seen in the lack of explicit mention of, e.g., "Recursive Speculative Decoding: Accelerating LLM Inference via Sampling Without Replacement" or similar tree/tree-of-N approaches, which could address the same efficiency-quality tradeoff. Including these could strengthen the empirical section.

2. Scope of Generalization Claims: Although Figure 4 and several model series are included, experiments mostly focus on models up to 14B parameters and a limited set of benchmarks. Claims about scalability to substantially larger backbones (hundreds of billions of parameters) or broader domains (e.g., multilingual or multimodal tasks) remain unproven, and the discussion in Appendix C defers these questions to future work.

3. Selection and Tuning of $\delta$ (Divergence Constraint): The hyperparameter $\delta$ critically determines the quality-throughput tradeoff. While the mechanism for grid search and choice is described, no principled or automated selection strategy is suggested, nor is the sensitivity to $\delta$ analyzed in detail beyond grid-search plots (see Figure 2). This may be a practical barrier for deployment, especially for practitioners unfamiliar with the KL/gamma tradeoff mechanics.

**Questions:**

1. The method currently relies on manual or grid-based tuning for the divergence parameter $\delta$. Is there a theoretically principled or adaptive way to set or anneal $\delta$ based on runtime feedback (e.g., based on target throughput or accuracy)? What would be the pitfalls in automating this choice?

2. While Cactus aims to bound divergence and preserve verifier fidelity, do any empirical failure cases show semantic or correctness failures? For example, what types of errors arise in GSM8K or GPQA when $\delta$ is set too high?

3. How would the approach scale to extremely large verifiers (e.g., >100B parameters) with draft models that are structurally or statistically misaligned? Does the performance gain or accuracy gap increase as model size scales up?

4. Related to Figure 1 and Table 1: In specific cases (e.g., GPQA with Cactus 1.0), Cactus surpasses the verifier's accuracy. Is this genuine improvement or due to variance/noise in evaluation or artifacts of metric normalization?

5. Could the framework be generalized to allow for multi-token acceptance rules (e.g., chains or blocks of tokens) beyond per-token KL constraint, and what would the mathematical/algorithmic implications be?

---

> ### Author Response · Authors · 2025-11-19
> **Official Comment by Authors (1/3)**
>
> We thank the reviewer for highlighting the theoretical rigor and practical impact of our method.
>
> > Empirical Baseline Coverage and Ablations: While the paper includes SpS, TAS, Top-k, Mentored decoding, and SpecCas as baselines, it does not systematically evaluate against very recent tree-based decoding advancements (such as recursive or blockwise candidate strategies). This limits direct benchmarking of Cactus's relative value in the face of alternative parallel or hierarchical sampling frameworks. The omission can be seen in the lack of explicit mention of, e.g., "Recursive Speculative Decoding: Accelerating LLM Inference via Sampling Without Replacement" or similar tree/tree-of-N approaches, which could address the same efficiency-quality tradeoff. Including these could strengthen the empirical section.
>
> Thanks for the comment. The scope of our paper mainly focuses on proposing a new acceptance criterion as an alternative to SpS or TAS, which was used as the underlying acceptance function in the recursive accepting phase of the suggested paper. As a result, we believe Cactus can be naturally extended to tree-based decoding with recursion or blocks. We have added the suggested work and enhanced the discussion on this in the related work section .
>
> > Scope of Generalization Claims: Although Figure 4 and several model series are included, experiments mostly focus on models up to 14B parameters and a limited set of benchmarks. Claims about scalability to substantially larger backbones (hundreds of billions of parameters) or broader domains (e.g., multilingual or multimodal tasks) remain unproven, and the discussion in Appendix C defers these questions to future work.
> > How would the approach scale to extremely large verifiers (e.g., >100B parameters) with draft models that are structurally or statistically misaligned? Does the performance gain or accuracy gap increase as model size scales up?
>
> Thanks for the suggestion. We would like to conduct further experiments with large models (up to 32B due to limited resources) on more benchmarks. However, these experiments would take more time. We would report back once we gather the results.
>
> > Selection and Tuning of  (Divergence Constraint): The hyperparameter  critically determines the quality-throughput tradeoff. While the mechanism for grid search and choice is described, no principled or automated selection strategy is suggested, nor is the sensitivity to  analyzed in detail beyond grid-search plots (see Figure 2). This may be a practical barrier for deployment, especially for practitioners unfamiliar with the KL/gamma tradeoff mechanics.
> > The method currently relies on manual or grid-based tuning for the divergence parameter . Is there a theoretically principled or adaptive way to set or anneal  based on runtime feedback (e.g., based on target throughput or accuracy)? What would be the pitfalls in automating this choice?
>
> Thanks for  the insightful comment. Given the nature of quality--efficiency trade-off, the suitable $\delta$ largely depends on what users prioritize.  However, Figure 2 shows that it is not a sensitive hyperparameter, and a range of 0-10 works generally fine. Based on our experiments across different models and tasks, we find that choosing $\delta=1$ largely maintains the performance of the verifier model while accelerating the decoding.

---

> > ### Author Response · Authors · 2025-11-19
> > **Official Comment by Authors (2/3)**
> >
> > > While Cactus aims to bound divergence and preserve verifier fidelity, do any empirical failure cases show semantic or correctness failures? For example, what types of errors arise in GSM8K or GPQA when  is set too high?
> >
> > Yes. We observed that the performance drops when $\delta$ is large, especially in that the reasoning makes less sense. Here is a case study:
> >
> > Question
> >
> > > `Question: Janet’s ducks lay 16 eggs per day. She eats three for breakfast every morning and bakes muffins for her friends every day with four. She sells the remainder at the farmers' market daily for $2 per fresh duck egg. How much in dollars does she make every day at the farmers' market?`
> >
> > $\delta=1$
> >
> > > `<think>\nOkay, let me try to figure out how much Janet makes each day at the farmers' market. Hmm, so the problem says she has ducks that lay 16 eggs per day. Then she eats three for breakfast and bakes muffins with four eggs each day. Then she sells the remaining eggs for $2 each. Alright, let's break this down step by step.\n\nFirst, total eggs per day: 16 eggs. Then she uses some for breakfast and baking. Let's see, for breakfast, she eats 3 eggs. Then for baking, she uses 4 eggs. So total eggs used each day would be 3 + 4 = 7 eggs. Therefore, the remaining eggs would be 16 - 7 = 9 eggs. Then she sells those 9 eggs at $2 each. So, how much does she make? 9 eggs multiplied by $2, which is 18 dollars.\n\nWait, let me check again. If she uses 3 eggs for breakfast and 4 for baking, that's 7. 16 minus 7 is indeed 9. Then 9 times 2 is 18. Yeah, that seems right. So her daily earnings would be $18. Hmm, is there any possibility I made a mistake here? Maybe I should re-read the problem to make sure I didn't miss anything.\n\nThe problem says she lays 16 eggs per day. Then she eats 3 for breakfast and bakes muffins each day with four. So the total used is 3+4=7. Subtract that from 16, which leaves 9. Multiply by $2 for selling, 9*2=18. Yep, that's correct. I think that's the answer.\n</think>\n\nJanet’s ducks lay 16 eggs per day. She uses 3 eggs for breakfast and 4 eggs for baking muffins, totaling 7 eggs used. The remaining eggs are $16 - 7 = 9$. She sells these at $2 per egg, earning $9 \\times 2 = 18$.  \n\n#### 18`
> >
> > $\delta=10^6$
> >
> > > `<think>\nOkay, so let\'s try to figure out how much Janet makes every day at the farmers\' market. Let me start by breaking down the problem step by step.` *[lengthy CoT omitted]* `so total eggs used is 4 * 4 = 16. Then, after eating 3 for breakfast, leftover is 16 - 3 = 13, then she sells them at 2 each: 13 * 2 = $26. \n\nYes, that seems to make sense. Therefore, the answer should be 26. But let me check if the question is asking "how much money does she make every day at the farmers\' market?" So yes, selling the remainder at $2 per egg. Therefore, answer is $26. \n\nTherefore, the answer is 26, which in the previous problems they had 5. This is a different case. So I think that\'s correct.\n</think>\n\nAnswer: 26\n\nThe problem states that Janet\'s ducks lay 16 eggs per day. She eats 3 eggs for breakfast and 4 eggs to make muffins daily. After these expenses, the remaining eggs are $ 16 - 3 - 4 = 9 $, which are sold at $2 per egg. Therefore, the total money she earns at the farmers\' market is $ 9 \\times 2 = 18 $. \n\nHowever, the phrase "bakes muffins for her friends every day with four" might imply that she uses 4 eggs per muffin, meaning she makes 4 muffins. If she uses 4 eggs each muffin, the total eggs consumed would be $ 4 \\times 4 = 16 $. Since she eats 3, the remaining eggs are $ 16 - 3 = 13 $, which are sold for $ 13 \\times 2 = 26 $. Hence, the correct answer is **26**. \n\n#### 26`
> >
> > From the above case study, we can see the reasoning is poor and lengthy when $\delta$ is large (more divergence allowed). Consequently, the result is wrong due to the low-quality chain-of-thought. We’ve presented the case study in the revision.
> >
> > > Related to Figure 1 and Table 1: In specific cases (e.g., GPQA with Cactus 1.0), Cactus surpasses the verifier's accuracy. Is this genuine improvement or due to variance/noise in evaluation or artifacts of metric normalization?
> >
> > We believe that it is a genuine improvement because (1) Cactus is often better by 2 sigma, and (2) the improvement is consistent across runs. Our hypothesis is that Cactus enables a “healthy” ensemble  of two models, which performs better than one model. We’re happy to explore ensemble effects in speculative decoding as future work (see discussion in Appendix C of the revision).

---

> > > ### Author Response · Authors · 2025-11-19
> > > **Official Comment by Authors (3/3)**
> > >
> > > > Could the framework be generalized to allow for multi-token acceptance rules (e.g., chains or blocks of tokens) beyond per-token KL constraint, and what would the mathematical/algorithmic implications be?
> > >
> > > Thanks for the insightful suggestion. Yes, we believe Cactus can be extended to multi-token acceptance rules given that it is an alternative to the SpS acceptance function. Therefore, one straightforward way is directly replacing the SpS/TAS acceptance functions in these methods with Cactus. However, we’re also exploring more integrated algorithms that jointly consider multiple steps as future work.
> > >
> > > ---
> > >
> > > **Summary**
> > >
> > > We thank the reviewer again for supporting our submission and providing detailed comments. In our response, we (1) explained practical choices and potential adaptive strategies for $\delta$, (2) discussed scalability to larger and more diverse models, and (3) clarified how Cactus complements recent tree-based / recursive decoding as a drop-in acceptance rule. We hope these clarifications address the reviewer’s concerns. We’re looking forward to your stronger support! Thank you.

---

> > > > ### Author Response · Authors · 2025-11-25
> > > > **Follow-up response with additional experiments**
> > > >
> > > > > Scope of Generalization Claims: Although Figure 4 and several model series are included, experiments mostly focus on models up to 14B parameters and a limited set of benchmarks. Claims about scalability to substantially larger backbones (hundreds of billions of parameters) or broader domains (e.g., multilingual or multimodal tasks) remain unproven, and the discussion in Appendix C defers these questions to future work.
> > > > > How would the approach scale to extremely large verifiers (e.g., >100B parameters) with draft models that are structurally or statistically misaligned? Does the performance gain or accuracy gap increase as model size scales up?
> > > >
> > > > Following our previous response, we have been conducting additional experiments with larger models, and we just obtained the new results.
> > > >
> > > > |        | GSM8K     | GSM8K    | IFEval    | IFEval   | GPQA      | GPQA     |
> > > > |--------|-----------|----------|-----------|----------|-----------|----------|
> > > > |        | Score↑    | AL↑      | Score↑    | AL↑      | Score↑    | AL↑      |
> > > > | SpS    | **95.30** | 5.03     | 83.36     | 2.61     | 40.40     | 3.73     |
> > > > | TAS    | 94.10     | 7.02     | 83.73     | 4.16     | 40.40     | 6.12     |
> > > > | Ours ($\delta=1$) | 94.40     | **7.13** | **85.21** | **4.47** | **41.92** | **6.36** |
> > > >
> > > > These experiments were conducted using the Qwen3 1.7B + 32B model pair. As shown in the table, Cactus not only has the longest acceptance length (AL) but also achieves the highest scores across 2 out of 3 tasks. This again confirms that the effectiveness of Cactus holds beyond smaller models and remains consistent even when both the drafter and the verifier are significantly larger.
> > > >
> > > > We hope our additional experiments further strengthen the empirical value of our work.

---

### Official Review · Reviewer_CzSq · 2025-10-31

**Soundness:** 3
**Presentation:** 3
**Contribution:** 3
**Rating:** 6
**Confidence:** 3

**Summary:**

This paper proposes Cactus, a novel speculative sampling method that provides explicit control over divergence from the verifier’s distribution to improve the token acceptance rate. It formulates speculative sampling as a constrained optimization problem that maximizes the draft token acceptance probability while keeping the target distribution within a bounded divergence from the verifier. Empirical results across multiple models and datasets show that Cactus achieves higher acceptance rates, often with improved accuracy compared to baseline methods.

**Strengths:**

1. Proposes principled trade-off between efficiency and fidelity. The paper proposes viewing speculative sampling as choosing an appropriate distribution $h$ that achieves a desired trade-off between draft token acceptance rate and divergence from the verifier. This view allows methods including SpS and TAS to be viewed in a unified framework.

2. Practical implementation. Based on the constrained optimization formulation, the paper derives a closed-form solution with cost comparable to SpS.

3. Empirical evaluations. Experiments on multiple models and datasets show improved acceptance length compared to SpS and TAS.

**Weaknesses:**

1. Missing evaluations on drafter quality. Qwen 3 models are used as both the verifier and the drafter. Since these models belong to the same family, their output distributions are likely to be similar, making it unclear how the results would change with different levels of inherent divergence between the verifier and drafter. Also, how does performance vary when using drafters of different sizes?

2. Missing evaluations on interactions with temperature. For the main experiments, one set values for top-p, top-k, and temperature (fairly low) is used. Further analysis on how different methods interact with sampling hyperparameters would be useful since the focus of the paper is on speculative $\textit{sampling}$.

3. Discussion on approximation validity. Cacus leverages second-order Taylor approximation assuming $\delta$ remains small. It is unclear whether $\delta$ values such as 0.75 justifies the approximation.

**Questions:**

Q. What are some possible explanations for the often better accuracies?

Q. The anonymous link to code seems broken?

---

> ### Author Response · Authors · 2025-11-19
>
> We are grateful to the reviewer for the careful reading and encouraging assessment. We are pleased that you found our method principled and recognize the practical implementation of Cactus.
>
>
> > Missing evaluations on drafter quality. Qwen 3 models are used as both the verifier and the drafter. Since these models belong to the same family, their output distributions are likely to be similar, making it unclear how the results would change with different levels of inherent divergence between the verifier and drafter. Also, how does performance vary when using drafters of different sizes?
>
> In terms of model family, speculative sampling methods typically use the drafter and verifier from the same family [1, 2, 3], which is also our setup. This is because the draft and verifier models should have the same tokenization to evaluate the probability of the tokens. Mapping different tokenizations is an interesting research direction on its own, which is beyond the scope of our paper.
>
> In terms of different draft model sizes, we provide the additional results as below
>
> | Draft size | Score↑ | AL↑  | Rej↓ |
> |-------|--------|------|------|
> | Verifier | 91.71 | - | - |
> | 0.6B  | 93.10  | 5.44 | -32% |
> | 1.7B  | 92.50  | 6.78 | -60% |
> | 4B    | 92.57  | 7.76 | -76% |
>
> Here, we use Qwen 3 14B as the verifier with different sizes of draft models from 0.6B to 4B with $\delta=1$ on GSM8K. The results show that using a larger draft model substantially improves the average acceptance length and reduces the token rejection rate. Importantly, our method consistently outperforms the naive decoding baseline across all configurations. These findings suggest that Cactus is robust and compatible with a wide range of draft model sizes. We will incorporate the complete results on other datasets in the revised manuscript.
>
> > Missing evaluations on interactions with temperature. For the main experiments, one set values for top-p, top-k, and temperature (fairly low) is used. Further analysis on how different methods interact with sampling hyperparameters would be useful since the focus of the paper is on speculative .
> > Discussion on approximation validity. Cacus leverages second-order Taylor approximation assuming  remains small. It is unclear whether  values such as 0.75 justifies the approximation.
>
> In the paper, we have provided results without top-$p$ and top-$k$ in Figure 2. The temperature is set to 1. This allows us to demonstrate the quality-efficiency tradeoff by setting different $\delta$ values. We hope the ablation study answers the reviewer’s question.
>
> > Q. What are some possible explanations for the often better accuracies?
>
> Thanks for asking the question. We hypothesize that it’s an ensemble effect, where a “healthy” mix of two models could perform better than one model. We’ve provided additional discussion in Appendix C in the revision.
>
> > Q. The anonymous link to code seems broken?
>
> We apologize for the issue. We’ve fixed it and our code is available at https://anonymous.4open.science/r/Cactus-2E4D/
>
> ---
>
> **Summary**
>
> Thank you again for the thoughtful comments and for pointing out several ways to strengthen our work. In the response, we’ve addressed all the points including conducting additional experiments. We hope our response addresses the reviewer’s concerns.

---

### Official Review · Reviewer_Qb8d · 2025-11-03

**Soundness:** 2
**Presentation:** 2
**Contribution:** 2
**Rating:** 2
**Confidence:** 3

**Summary:**

The core idea—casting speculative decoding as a constrained optimization and applying a KL-divergence bound with a second-order Taylor approximation—appears as an incremental re-interpretation of existing acceptance strategies rather than a fundamentally new algorithmic principle. Theoretical development largely repackages known mechanisms, while empirical gains emphasize acceptance/throughput and depend heavily on the hyperparameter δ; accuracy robustness improvements are inconsistent across tasks. Overall, the contribution does not clear the bar for novelty and impact.

The manuscript exhibits noticeable typesetting glitches (e.g., unnecessary hyphenation and line breaks in the title/long tokens, inconsistent styles for algorithms/tables/captions, and minor inconsistencies in notation and reference formatting. These issues hurt readability and professional polish; a thorough pass against the official template is needed.

**Strengths:**

The work introduced a principled, divergence-controlled acceptance rule (Cactus) that balances fidelity and efficienc.

**Weaknesses:**

The manuscript exhibits noticeable typesetting glitches (e.g., unnecessary hyphenation and line breaks in the title/long tokens, inconsistent styles for algorithms/tables/captions, and minor inconsistencies in notation and reference formatting. These issues hurt readability and professional polish; a thorough pass against the official template is needed.

**Questions:**

The manuscript exhibits noticeable typesetting glitches (e.g., unnecessary hyphenation and line breaks in the title/long tokens, inconsistent styles for algorithms/tables/captions, and minor inconsistencies in notation and reference formatting. These issues hurt readability and professional polish; a thorough pass against the official template is needed.

---

> ### Author Response · Authors · 2025-11-19
>
> Thanks for the comments.
>
> > The manuscript exhibits noticeable typesetting glitches (e.g., unnecessary hyphenation and line breaks in the title/long tokens, inconsistent styles for algorithms/tables/captions, and minor inconsistencies in notation and reference formatting. These issues hurt readability and professional polish; a thorough pass against the official template is needed.
>
> Our template strictly follows the ICLR 2026 template. The hyphenation and line breaks are automatically generated by the LaTeX style file. On the contrary, all other reviewers rate our presentation score as 3. Reviewers sKn6 and 6cgh explicitly highlighted that our paper is well-written and highly readable.
> That being said, the formatting style does not hurt our central scientific contributions. **We urge the reviewer to revisit our paper and reassess our work accordingly**.

---

### Official Review · Reviewer_6cgh · 2025-11-03

**Soundness:** 2
**Presentation:** 3
**Contribution:** 2
**Rating:** 4
**Confidence:** 3

**Summary:**

This paper introduces Cactus, a novel speculative sampling method for accelerating auto-regressive decoding in large language models (LLMs). Building on prior work like vanilla speculative sampling (SpS) and typical acceptance sampling (TAS), the authors reformulate speculative sampling as a constrained optimization problem to trade off acceptance rates against divergence from the verifier model's distribution. They derive a closed-form approximation using KL divergence and provide theoretical guarantees on bounded divergence. Empirical evaluations on benchmarks like GSM8K, IFEval, and GPQA using models from Qwen, Gemma, R1, and LLaMA series show that Cactus achieves higher acceptance rates and throughput than baselines while preserving or improving output quality.

**Strengths:**

1. The paper is well-written and highly readable, with the authors providing a detailed introduction to the proposed method, along with in-depth theoretical contributions and proofs.
2. The "training-free" nature and minimal computational overhead of Cactus make it extremely easy to integrate into existing inference frameworks, endowing it with high practical value.
3. The experiments cover a variety of benchmark datasets as well as multiple model families and sizes. The results consistently show that Cactus outperforms SpS and TAS in terms of acceptance length and rejection rate, while maintaining stable accuracy.

**Weaknesses:**

1. It only includes three types of datasets: mathematical reasoning, instruction following, and scientific question answering, and lacks experimental results on Spec-Bench [1]—a more commonly used benchmark for speculative decoding. Spec-Bench covers multiple tasks such as multi-turn dialogue, translation, mathematical reasoning and so on.
2. The experiments only cover small-to-medium-sized models (with a maximum of 14B parameters) and do not involve larger models (e.g., 70B parameters) where memory bottlenecks are more prominent.
3. The baselines are relatively outdated. Key baseline methods such as Medusa [2] and EAGLE-2/3 [3, 4] are not comprehensively compared in the main results.
4. The code in the anonymous link (https://anonymous.4open.science/r/Cactus-2E4D/) is empty.

[1] Xia H, Yang Z, Dong Q, et al. Unlocking Efficiency in Large Language Model Inference: A Comprehensive Survey of Speculative Decoding [C]. Findings of the Association for Computational Linguistics ACL 2024. 2024: 7655-7671.

[2] Cai T, Li Y, Geng Z, et al. Medusa: Simple LLM Inference Acceleration Framework with Multiple Decoding Heads [C]. International Conference on Machine Learning. PMLR, 2024: 5209-5235.

[3] Li Y, Wei F, Zhang C, et al. EAGLE-2: Faster Inference of Language Models with Dynamic Draft Trees [C]. Proceedings of the 2024 Conference on Empirical Methods in Natural Language Processing. 2024: 7421-7432.

[4] Li Y, Wei F, Zhang C, et al. Eagle-3: Scaling up inference acceleration of large language models via training-time test [J]. arXiv preprint arXiv:2503.01840, 2025.

**Questions:**

1. How sensitive is Cactus to δ? Can the authors provide ablation experiments?
2. How to select the optimal δ? And how should δ be chosen for different tasks?

---

> ### Author Response · Authors · 2025-11-19
>
> We thank the reviewer for the careful and constructive assessment. We are glad that the reviewer appreciated the training-free design and high practical value of Cactus.
>
> > It only includes three types of datasets: mathematical reasoning, instruction following, and scientific question answering, and lacks experimental results on Spec-Bench [1]—a more commonly used benchmark for speculative decoding. Spec-Bench covers multiple tasks such as multi-turn dialogue, translation, mathematical reasoning and so on.
>
> We thank the reviewer for suggesting SpecBench. The benchmarks in our paper (GSM8K, IFEval, and GPQA diamond) are extensively used to evaluate modern LLMs [1, 2, 3]. In fact, SpecBench (suggested by the reviewer) partially overlaps with our evaluation as it samples from GSM8K.
>
>
> That being said, we’re happy to conduct additional experiments on SpecBench, but it might take additional time for us to implement our method following the SpecBench’s protocol. We would report back once we gather the results.
>
> > The experiments only cover small-to-medium-sized models (with a maximum of 14B parameters) and do not involve larger models (e.g., 70B parameters) where memory bottlenecks are more prominent.
>
> Thanks for the suggestion. Again, conducting experiments on large-scale LLMs would take more time. We’re running a 32B Qwen 3 model at this moment (as 70B is too big for us). We’ll update the results when we have them.
>
> > The baselines are relatively outdated. Key baseline methods such as Medusa [2] and EAGLE-2/3 [3, 4] are not comprehensively compared in the main results.
>
> We would like to clarify that Medusa and Eagle series are trained models, whereas our proposed method is an acceptance criterion without any training. In the paper, we have discussed this difference in Section 4. In our experiments, extensively compared with SpS and TAS, the underlying acceptance criteria used in Eagles and Medusa models. Given the scope of this paper, we leave the extension of Cactus to trainable tree-like attention modules to future work.
>
> > The code in the anonymous link (https://anonymous.4open.science/r/Cactus-2E4D/) is empty.
>
> We apologize for the inconvenience. We have resolved the technical glitch and the repo should be viewable now.
>
> > How sensitive is Cactus to δ? Can the authors provide ablation experiments?
> > How to select the optimal δ? And how should δ be chosen for different tasks?
>
> Thanks for asking the questions. In Section 3.3, we conducted $\delta$ sweeping from 1 to 1e6. We empirically observe that $\delta \approx 1$ usually yields stable text quality while largely accelerating the inference speed.
>
>
> ---
>
> **Summary**
>
>
> We again thank the reviewer for the thoughtful feedback and for pointing out several directions to further strengthen the work. We have clarified on every concern raised by the reviewer, and will report additional experiments once we gather the results.
>
>
> [1] Qwen3 Technical Report. arXiv 2505.09388.\
> [2] The Llama 3 Herd of Models. arXiv 2407.21783. \
> [3] DeepSeek-V3 Technical Report. arXiv 2412.19437.

---

> > ### Author Response · Authors · 2025-11-25
> > **Follow-up response with additional experiments**
> >
> > > It only includes three types of datasets: mathematical reasoning, instruction following, and scientific question answering, and lacks experimental results on Spec-Bench [1]—a more commonly used benchmark for speculative decoding. Spec-Bench covers multiple tasks such as multi-turn dialogue, translation, mathematical reasoning and so on.
> >
> > Following our previous response, we have now gathered the experiment results on Spec Bench as below.
> >
> >
> > |        | MT Bench | Translation | Summarization | QA    | Math  | RAG   | Accepted | Overall |
> > |--------|----------|-------------|---------------|-------|-------|-------|---|---------|
> > | SpS    | 2.01x    | 1.40x       | 1.92x         | 1.85x | 1.83x | 1.86x | 3.20 | 1.81x   |
> > | Ours ($\delta=1$) | 2.09x    | 1.40x       | 2.04x         | 1.95x | 1.86  | 1.92x | 3.29 | 1.88x   |
> >
> > In this experiment, the model combination is Qwen3 0.6B + 14B with a temperature of 0.6. Cactus is benchmarked without hyper-parameter tuning (i.e., $\delta=1$). It is evident that our method provides immediate acceleration over SpS without any training required. Across different domains, Cactus is consistently faster than SpS. Overall, Cactus achieves a +88% speed gain over the baseline, significantly reducing compute costs. Note that the speeds above are measured using the HuggingFace transformers framework, which is less optimized. The real-world performance gain could be even larger with more optimization (as justified by our other experiments using VLLM).
> >
> > > The experiments only cover small-to-medium-sized models (with a maximum of 14B parameters) and do not involve larger models (e.g., 70B parameters) where memory bottlenecks are more prominent.
> >
> > Following our previous response, we have also finished the experiments with larger models.
> >
> > |        | GSM8K     | GSM8K    | IFEval    | IFEval   | GPQA      | GPQA     |
> > |--------|-----------|----------|-----------|----------|-----------|----------|
> > |        | Score↑    | AL↑      | Score↑    | AL↑      | Score↑    | AL↑      |
> > | SpS    | **95.30** | 5.03     | 83.36     | 2.61     | 40.40     | 3.73     |
> > | TAS    | 94.10     | 7.02     | 83.73     | 4.16     | 40.40     | 6.12     |
> > | Ours ($\delta=1$) | 94.40     | **7.13** | **85.21** | **4.47** | **41.92** | **6.36** |
> >
> > These experiments were conducted using the Qwen3 1.7B + 32B model pair. As shown in the table, Cactus not only has the longest acceptance length (AL) but also achieves the highest scores across 2 out of 3 tasks. This again confirms that the effectiveness of Cactus holds beyond smaller models and remains consistent even when both the drafter and the verifier are significantly larger.
> >
> >
> > ---
> >
> > **Summary**
> >
> > We have completed the additional experiments as promised in our previous response. The results are now included in the appendix with proper citations. We believe these results provide strong evidence of the effectiveness of Cactus. We hope that all previous concerns have now been addressed, and we respectfully invite the reviewer to raise the score in light of our response. Please do not hesitate to let us know if you have any further questions. Thank you for your time and effort!

---

### Official Review · Reviewer_sKn6 · 2025-11-04

**Soundness:** 3
**Presentation:** 3
**Contribution:** 3
**Rating:** 6
**Confidence:** 3

**Summary:**

This paper proposes CACTUS, a novel constrained acceptance speculative sampling algorithm that accelerates large language model decoding while maintaining output fidelity. The key idea is to cast speculative sampling as a constrained optimization problem that explicitly trades off divergence from the verifier distribution against acceptance rate. By leveraging an f-divergence–bounded objective, the authors derive a closed-form solution (with KL divergence) that ensures provably controlled deviation from the verifier while allowing more draft tokens to be accepted. Extensive experiments on multiple model families (Qwen, Gemma, R1, LLaMA) and benchmarks (GSM8K, IFEval, GPQA) show that CACTUS consistently improves throughput and acceptance length without hurting accuracy. Compared with typical acceptance sampling (TAS), it demonstrates both higher acceptance rates and more stable quality, validating its theoretical motivation.

**Strengths:**

The paper is well-written. The proposed approach is novel. The evaluation is comprehensive and analysis is in-depth.

**Weaknesses:**

1. The observed improvements of CACTUS appear to correlate strongly with the capability of the underlying LLMs. For instance, larger verifiers (e.g., 14B) consistently outperform smaller ones (e.g., 8B), and reasoning-tuned models achieve higher acceptance and throughput than non-reasoning counterparts. This raises the question of whether the proposed method inherently benefits from the stronger self-consistency and error-correction capacity of powerful models, rather than from the constrained optimization formulation itself. A more detailed analysis or discussion of this correlation would strengthen the empirical claims.
2. The evaluation primarily reports acceptance length and rejection ratio as proxies for throughput, but omits key metrics such as average generation length or tokens per completion. Without these, it is difficult to fully interpret both the quality improvements and the real speedup.

**Questions:**

See above

---

> ### Author Response · Authors · 2025-11-19
>
> We thank the reviewer for the thoughtful feedback and positive assessment of our work. We’re encouraged that you say our paper is “well-written”, our “proposed approach is novel”, and our “evaluation is comprehensive”.
>
>
> > The observed improvements of CACTUS appear to correlate strongly with the capability of the underlying LLMs. For instance, larger verifiers (e.g., 14B) consistently outperform smaller ones (e.g., 8B), and reasoning-tuned models achieve higher acceptance and throughput than non-reasoning counterparts. This raises the question of whether the proposed method inherently benefits from the stronger self-consistency and error-correction capacity of powerful models, rather than from the constrained optimization formulation itself. A more detailed analysis or discussion of this correlation would strengthen the empirical claims.
>
> We appreciate the reviewer's observation on the correlation between LLM capabilities and Cactus performance. This is expected because Cactus is designed to retain the performance of the verifier model with our constrained optimization framework. The correlation in fact confirms the effectiveness of our method.
>
> On the other hand, the acceptance rate could be higher when the draft and verifier models are more similar for all methods (including Cactus). We additionally conduct the following results to confirm that our method works across different selections of draft-verifier pairs:
> | Draft size | Score↑ | AL↑  | Rej↓ |
> |-------|--------|------|------|
> | Verifier | 91.71 | - | - |
> | 0.6B  | 93.10  | 5.44 | -32% |
> | 1.7B  | 92.50  | 6.78 | -60% |
> | 4B    | 92.57  | 7.76 | -76% |
>
> Here, we use Qwen 3 14B as the verifier with different sizes of draft models from 0.6B to 4B with $\delta=1$ on GSM8K. The results show that using a larger draft model substantially improves the average acceptance length and reduces the token rejection rate. Importantly, our method consistently outperforms the naive decoding baseline across all configurations in terms of efficiency. These findings suggest that Cactus is robust and compatible with a wide range of draft model sizes. We will incorporate the complete results with other datasets in the revised manuscript.
>
> > The evaluation primarily reports acceptance length and rejection ratio as proxies for throughput, but omits key metrics such as average generation length or tokens per completion. Without these, it is difficult to fully interpret both the quality improvements and the real speedup.
>
> We thank the reviewer for asking the question. We provide additional results on average generation length (counting CoT thinking tokens) on each task as below
>
> |Method | GSM8K | IFEval | GPQA |
> |-------|--------|-------|--------|
> | SpS | 588.2 | 585.3 | 2484.6 |
> | TAS | 591.3 |  670.2 |  2561.3 |
> | Cactus | 608.1 | 632.3 | 2544.4 |
>
> Here, we used the 0.6B+14B model pair. We see that all methods have a similar generation length, which is expected because our method is designed to have a similar distribution. This again shows the effectiveness of our method.
>
> In addition, the generation length is not a metric of efficiency, given that different tasks require different lengths to complete (e.g. GPQA completions are much longer). In this paper, we mainly reported average acceptance length, rejected tokens, and throughput (wallclock time). They comprehensively evaluate the generation efficiency from different perspectives. As shown in Table 1 and Figure 3, Cactus consistently outperforms baselines in different aspects of efficiency.
>
> ---
>
> **Summary**
>
> We again thank the reviewer for the constructive comments and careful evaluation of our submission. In this response, we have (1) analyzed the interaction between verifier/draft capability and Cactus with new experiments across draft sizes, and (2) reported average generation lengths while clarifying why acceptance length, rejected tokens, and wallclock throughput are used as efficiency indicators. We hope these clarifications address the concerns, and we would be grateful if the reviewer could raise the score in light of our responses. Thank you for your time and support!

---

### Meta-Review · Area_Chair_ooug · 2026-01-10

**Summary:**

The paper studies approximate speculative decoding (SD) through relaxing the acceptance criteria by solving a divergence constrained optimization problem. The paper complements the principled motivation with empirical evidence on a diverse set of benchmarks showing improved latency with neural quality on most datasets.

Modulo one reviewer with low-quality feedback, there is a positive consensus among the reviewers about the principled formulation of the problem and good empirical evidence. The reviewer raised concerns about the extensiveness of the experiments regarding the experimented models and benchmark dataset, as well as the robustness analysis on the hyperparameter choice, which are mostly addressed in the author's rebuttal.

One concern that the authors failed to address is that the paper doesn't contain experiments on more recent SD techniques such as EAGLE, which I think it is a nice experiment to have. However, I don't see this as a blocker for the paper's acceptance.

Overall, I would recommend the paper for acceptance. The authors should include the reviewers' suggestions to improve their final draft.

**Reviewer Concerns:**

See above discussion.

**Reviewer Scores:**

I think most reviewers will maintain their positive rating of the paper.

---

### Decision · Program_Chairs · 2026-01-26

Accept (Poster)